# Atomic force microscope protocols for characterising the elastoviscoplastic biomechanical properties of corneocytes
Ana S. Évora, Zhihua Zhang, Simon A. Johnson [ID], Zhibing Zhang & Michael J. Adams [ID] ✉

Corneocytes, the fundamental units of the epidermis outer layer, are essential for skin's barrier function. This study employs Atomic Force Microscopy (AFM) to explore the topographical and biomechanical properties of volar forearm cells. A detailed protocol is presented to eliminate experimental artefacts that have led to variability in reported Young's moduli. The goal is to create a consistent material model reflecting the elastic and inelastic behaviour of corneocytes. Using standard sharp AFM probes allows for accurate cell topography capture and targeted indentation for mechanical property measurements without changing probes. The methodology for interpreting mechanical data from sharp indenters is also addressed. Results indicate that corneocytes in a dry state exhibit Young's moduli similar to glassy organic polymers and demonstrate viscoplastic behaviour, described by the Herschel-Bulkley model. These detailed protocols enhance our understanding of skin biomechanics, potentially guiding advancements in biomimetic materials and dermatological studies.

Skin is the interface between an organism and the external environment, acting as the first line of defence against a range of external insults such as mechanical loadings[1,2]. The contribution of the deeper skin layers, such as the dermis, to the architecture and mechanical function of the skin has been studied extensively[3,4]. However, the role of the epidermis, and particularly that of the Stratum Corneum (SC), on the mechanical integrity of skin has been mostly ignored. Nevertheless, the SC is the most superficial layer of the skin, and thus it must adapt to the physiological and boundary requirements of each anatomical location, as exemplified by the special case of glabrous skin[5]. Consequently, it is important to explore to what extent the SC contributes to the mechanical resistance of skin across different body sites and under different conditions.

The stiffness and strength of any tissue depend on the mechanical properties of its constituent cells and extracellular matrix, rendering the mechanical study of single cells key to understanding many biological processes[6]. The SC, its main cell type (corneocytes) and lipids have been characterised topographically, mechanically, and chemically using Atomic Force Microscopy (AFM)[7–10]. The Young's modulus of corneocytes has been previously reported in the literature[8,9,11–13], but with a high degree of variability, ranging from a few MPa[12] up to 0.4 GPa[14]. Most of these studies consider that they deform elastically so that the loading and unloading curves can be described by applying the Hertz or JKR models, which are based on a spherically capped indenter[8,13]. Moreover, as demonstrated by Beard et al.[7], when indenting corneocytes, there is plastic deformation at the yield strain.

These authors performed mechanical tomography with nanoneedles and used the Oliver-Pharr analysis to calculate the elastic modulus from the unloading curves. The value of the Young's modulus was observed to increase with indentation depth, being about 100 MPa for indentation depths <50 nm, and 500 MPa for greater depths[7]. Milani et al.[8] also reported an increase in the modulus with indentation depth, being 250 and 300–520 MPa for indentation depths of <50 nm and >50 nm, respectively. They used a tomographic technique that was performed with pyramidal probes and relied on a linear elastic model based on a rigid cone indenting a flat surface. The corneocytes from the inner SC presented smaller Young's modulus values (250–300 MPa) compared to superficial cells (300–520 MPa)[8]. The main differences in the protocols of these two studies, apart from the contact mechanics models employed, were the geometries of the probes and the stiffnesses of the cantilevers. While Beard et al.[7] used cantilevers with a spring constant of 35 N/m, and cylindrical nanoneedles with a tip radius of 20 and 35 nm, Milani et al.[8] used more compliant cantilevers (2.5 N/m) and three-sided pyramidal probes. There were also differences in how the authors harvested the cells. Beard et al.[7] used hair removal wax melted on a microscope glass slide to strip the SC, as a relatively hard substrate for AFM. Milani et al.[8] stripped the SC using circular adhesive tapes (D-squame, CuDerm Corp., Dallas, TX, US) and adhered the tapes to a microscope glass slide using double-sided tape. The selection of the substrate is of considerable importance for nanoindentation, as will be discussed later; soft substrates may have a large compliance compared with the sample[15].

School of Chemical Engineering, University of Birmingham, Birmingham, UK. ✉e-mail: m.j.adams@bham.ac.uk

While the use of AFM as a nanoindentation technique has now been established, there are still many uncertainties in the protocols that must be addressed in order to achieve standard practices that allow comparisons across different studies. Such uncertainties range from intrinsic calibrations required by the system, including the spring constant of the cantilever, sensitivity of the system, and geometry of the probe, to the analysis of the data, including the appropriate selection of the contact mechanics model. One of the main uncertainties in most AFM nanoindentation studies is the overall geometry and tip radius of the probes, since most authors assume that the nominal values given by the manufacturer are accurate[8,12,14]. This is critical for selecting the correct contact mechanics model. Spherical colloidal probes are usually used for the mechanical analysis of cells due to their well-defined geometry and the need to avoid the non-linear response of the cell material that can result in cellular damage[15]. However, by using sharp probes for nanoindentation, it is possible to apply sufficient stresses that cause plastic deformation and thus allow the elastoplastic properties to be characterised for cells such as corneocytes, which are more resilient to such damage. Furthermore, one of the main advantages of AFM is the coupling of topographical and biomechanical analyses that can be achieved with standard imaging probes, also known as sharp AFM tips. This is particularly important for assessing the correlation of the surface properties of corneocytes with their mechanical behaviour. In fact, the presence of circular nano-objects at the surface of skin cells has been associated with certain skin conditions, such as atopic dermatitis[11], but there have not been any attempts to correlate these maturation properties with the stiffness of the skin cells.

Standard imaging AFM probes are usually assumed to have spherical tips and to behave either as pyramids or cones for indentations greater than the tip radius. Therefore, when assuming a purely linear elastic deformation (as in most studies of cells), the Hertzian model is selected for indentation depths, $h$, that are much smaller than the nominal tip radius, $R$. In this case, the tip geometry may be treated as parabolic and then $F \propto h^{3/2}$, where $F$ is the applied force. When $h > R$, it is often assumed that $F \propto h^2$ for a conical geometry[16] since the given nominal tip radius is usually quite small ($R < 10$ nm) using standard AFM probes. These assumptions have been challenged by Fuginami et al.[17] for the case of adhesive elastic contacts. They explored the cone-paraboloid transition for sharp AFM tips and showed that unsuitable geometric assumptions can give large errors. That is, it is important to know the total tip geometry, from the tip to a distance equal to the maximum contact depth, $h_c$.

Therefore, given the variability of reported values of the Young's modulus and the lack of a comprehensive protocol for measuring the mechanical properties of corneocytes using AFM, the current work aims to explore the main important variables of AFM nanoindentation, and to describe the appropriate data analysis for characterising the mechanical behaviour. The effective geometry of the AFM tip was determined by indenting a silicone elastomer reference material in the presence of an aqueous sodium dodecyl sulphate (SDS) solution to attenuate the adhesion.

From the tip geometry and nanoindentation unloading curves, it was then possible to calculate the elastic modulus of corneocytes in the dry state collected from three healthy subjects. The mechanical properties of the corneocytes were further studied by performing stress-relaxation experiments. It was observed that the cells exhibited viscoplastic behaviour that could be described by the Herschel-Bulkley material model.

## Results

### Initial calibration of the tip radius using Environmental Scanning Electron Microscopy (ESEM) and AFM tip characterisation tool

The approximate overall geometry of the probe used in the current work was obtained from Environmental Scanning Electron Microscopy (ESEM) images (Fig. 1a), which revealed an overall conical-like shape. For AFM, it is common to use Tapping Mode (TM) imaging of a reference sample to determine the tip geometry by applying a blind tip estimation Gwyddion algorithm[18] to obtain a 3D projection of the tip and a value for the tip radius[19,20]. This analysis resulted in an asymmetric paraboloid-like structure for the first 50 nm of the tip apex with a radius of curvature in all directions of $53 \pm 7$ nm (Fig. 1b).

### Nanoindentation of PDMS

To derive a more complete quantitative description of the probe geometry, indentations were performed on a reference elastomeric material, polydimethylsiloxane (PDMS)[21]. A complete description of the calibration methodology is described in Supplementary Material S1. The PDMS exhibited linear elastic behaviour for strains less than the maximum value corresponding to that applied to the indentation of the corneocytes (Supplementary Material S5). The surface topography and representative force data of the PDMS are presented in Fig. 2a, b, respectively.

The relationship between the geometry of a probe and the resulting indentation force as a function of indentation depth for PDMS can be obtained by first considering the general expression for the contact stiffness, $S$[16]. Assuming that an effective geometry may be obtained by treating the probe as axisymmetric and that it is rigid, $S$ may be expressed as follows:

$$S = \frac{dF}{dh} = \frac{2E}{1 - \nu^2} a = 2E^* a \tag{1}$$

where $F$ is the indentation force, $h$ is the indentation depth, and $a$ is the contact radius. $E$ and $\nu$ are the Young's modulus and Poisson's ratio of the PDMS, and $E^*$ is the plane strain elastic modulus; it was assumed that $\nu = 0.5$ since PDMS is considered to be incompressible.

The value of $E^*$ for the PDMS calibration specimen used was measured to be $3.79 \pm 0.21$ MPa using a micromanipulator equipped with a flat-ended cylindrical probe. This is routinely employed to determine the mechanical properties of micro-particles (Supplementary Material S5[22]). For the current purpose, the linear slope of the force as a function of the indentation depth

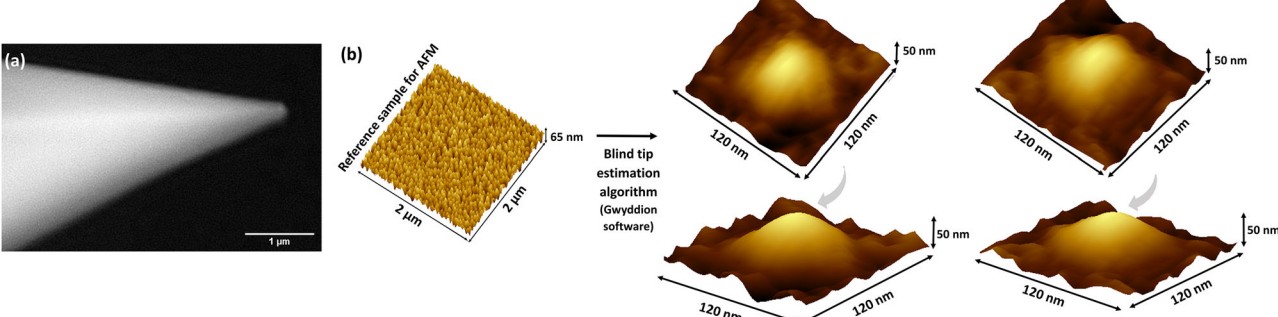

**Fig. 1 | Preliminary characterisation of the AFM probe geometry. a** ESEM image. Scale bar = 1 μm. **b** A blind tip estimation was performed using an AFM reference sample for which the measured surface topography is shown. The tip radius evolution was calculated during imaging using Gwyddion software (partial analysis) applied to 20 horizontal strips of the image. 3D projections of the estimated tip shape are shown from two viewing angles (180° apart).

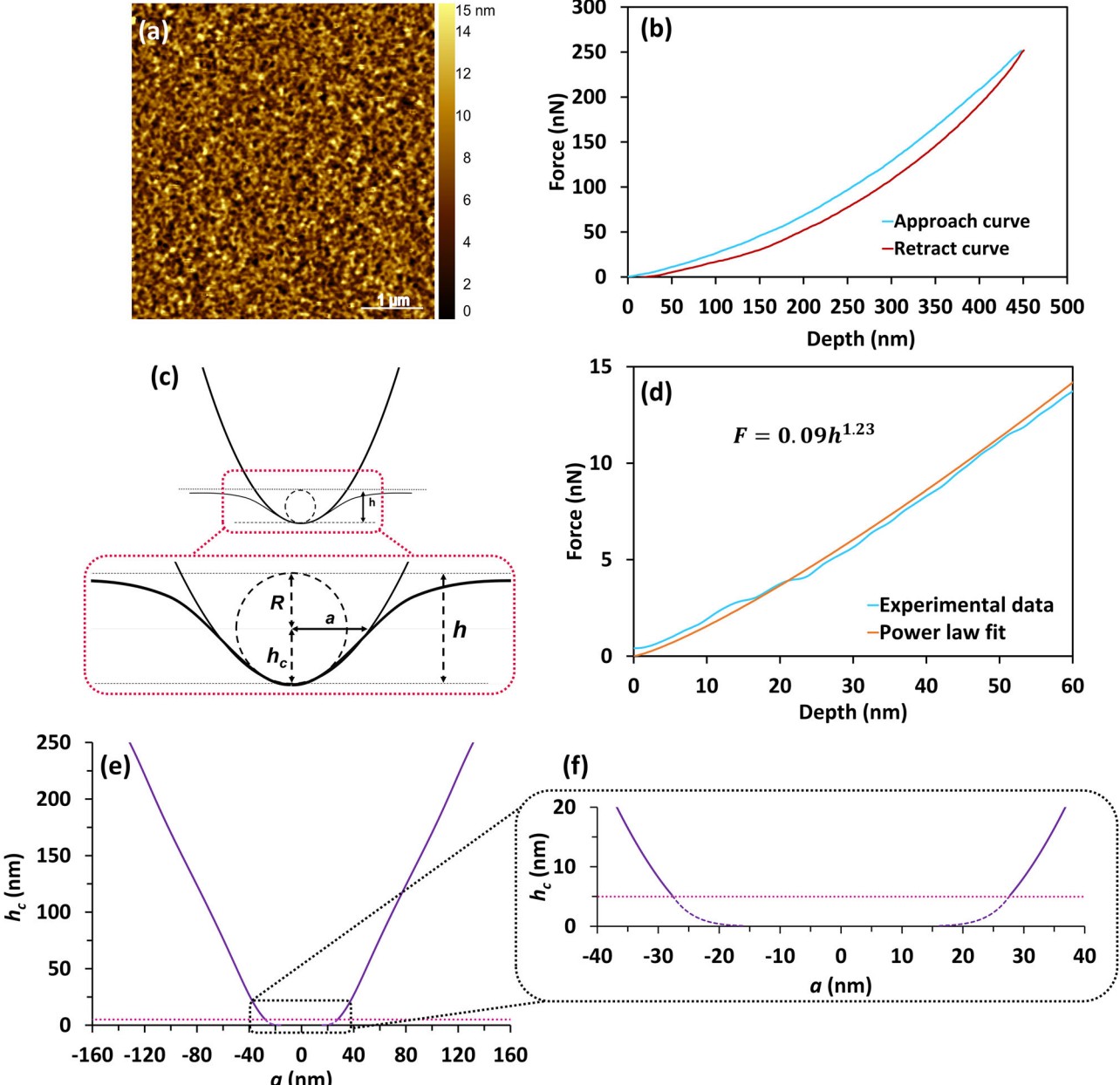

**Fig. 2 | AFM tip calibration using a PDMS reference elastomer. a** Surface topography of the PDMS sample (5:1) used for the characterisation of the probe. Scale bar = 1 μm. **b** Representative indentation loading and unloading curve for the PDMS in 0.1% aqueous SDS showing a typical elastic deformation with minimal adhesion. **c** Assumption of a hard standard TM-AFM tip indenting a soft elastic material as a paraboloid of revolution, where $h_c$ is the contact depth, $h$ is the total indentation depth, $R$ is the end radius of curvature of the parabola, and $a$ is the contact radius.

**d** Representative power law fit to the initial 60 nm of a PDMS loading curve presented in **b**. **e** Average tip radius function obtained from Eq. 10 (also see Supplementary Fig. S2). **f** Zoom-in of the tip geometry for a maximum contact depth of 20 nm. The tip geometry was determined from a minimum $h_c$ = 5 nm, as indicated by the horizontal dotted line. The dashed line shows the power law for $h_c$ = 5 nm, where $n = 7.55 \pm 0.20$ and $c = (7.90 \pm 3.90) \times 10^{-11}\,\mu m^{-6.55}$; see Eq. (6) and Supplementary Table S1.

gives the contact stiffness and the (constant) contact radius is known so that $E^*$ can be calculated from Eq. (1). In the case of the AFM nanoindentation experiments, $E^*$ is assumed to be given by the same value but the probe geometry is the unknown.

For indentations with a spherically tipped probe, the Hertzian force as a function of depth relationship is commonly used to obtain $E^*$ from the loading curve[23]:

$$F = \frac{4}{3}E^* R^{1/2} h^{3/2} \qquad (2)$$

where $R$ is the radius of the sphere.

Equation (2) can be obtained by integrating Eq. (1), after approximating the circular contact profile, $h_c(a)$, by a parabola (see Fig. 2c):

$$h_c = \frac{a^2}{2R} \qquad (3)$$

where $h_c$ is the contact depth, and by using the paraboloid relationship between the indentation depth and the contact depth[16]:

$$h = 2h_c \qquad (4)$$

**Table 1 | AFM tip geometry parameters obtained from an elastic analysis of PDMS indentation data**

| $h_c$ (nm) | 10 | 20 | 40 | 60 | 80 | 160 |
|---|---|---|---|---|---|---|
| $m$ | $1.184 \pm 0.004$ | $1.240 \pm 0.004$ | $1.311 \pm 0.002$ | $1.380 \pm 0.002$ | $1.45 \pm 0.002$ | $1.663 \pm 0.005$ |
| $n$ | $5.41 \pm 0.10$ | $4.62 \pm 0.08$ | $3.21 \pm 0.03$ | $2.63 \pm 0.01$ | $2.22 \pm 0.01$ | $1.51 \pm 0.01$ |
| $c$ ($\mu m^{1-n}$) | $(7.80 \pm 3.11) \times 10^{-8}$ | $(5.90 \pm 1.31) \times 10^{-6}$ | $(1.85 \pm 0.21) \times 10^{-4}$ | $(1.65 \pm 0.09) \times 10^{-3}$ | $(8.30 \pm 0.02) \times 10^{-3}$ | $0.16 \pm 0.01$ |

The load index, $m$, of the power law was calculated at different contact depths, $h_c$, based on a polynomial fit of PDMS loading curves. The plane strain elastic modulus, $E^*$, of the PDMS specimen required to derive the tip geometry was independently calculated from micromanipulation experiments based on the indentation of 3 different regions. The derived parameters $c$ and $n$ define the local geometry of the AFM tip at different selected contact depths. Mean $\pm$ 1 SD.

Equations (3) and (4) are a close approximation for a spherical indenter, as is Eq. (2), provided that the contact radius is small compared to the sphere radius. This is not the case for the probes used in the current AFM nanoindentation measurements (see Fig. 2c).

AFM loading curves can be fitted to a power law in order to investigate the applicability of Eq. (2):

$$F = bh^m \tag{5}$$

where $b$ and $m$ are the power law load coefficient and load index, respectively.

For PDMS, when analysing the AFM loading curves (Fig. 2b), it was found that the best-fit for the power law index was not 3/2 but varied with indentation depth, being slightly greater than 1 for fitting the data close to contact (Fig. 2d), and $\approx$1.7 for data at deeper depths. In fact, experimentally, a 2nd order polynomial with zero intercept appears to provide a more satisfactory fit over the entire range of the loading curve, rather than a single power law. This indicates that the probe geometry is more closely approximated by a truncated cone than a single power law, such as a paraboloid ($m = 1.5$).

Thus, the blind estimation of the tip shape is misleading. Consequently, in the current work, an alternative analysis was used to obtain a more accurate geometry of the probe. In summary, this is based on multi-term polynomial smoothing and averaging of multiple loading curves to calculate, firstly, the contact radius as a function of the indentation depth, $a(h)$, using Eq. (1) and, secondly, $m(h)$, the power law load index as a function of the indentation depth. The tip radius profile, $a(h_c)$, is derived from these two functions utilising indentation equations for a generalised power law indenter profile:

$$h_c = ca^n \tag{6}$$

where $c$ and $n$ are the power law indenter profile coefficient and indenter profile index, respectively. Equation (6) is the general form of Eq. (3) when $n = 2$, which is related to the index $m$ in Eq. (5) as follows[24]:

$$n = \frac{1}{m-1} \tag{7}$$

The relationship between the indentation depth and the contact depth is given by:

$$h = \kappa h_c \tag{8}$$

where $\kappa$ is a scaling factor given by[24]:

$$\kappa = \sqrt{\pi} \frac{\Gamma\left(\frac{n+2}{2}\right)}{\Gamma\left(\frac{n+1}{2}\right)} \tag{9}$$

where $\Gamma(.)$ is the Gamma function. Equation (8) is the general form of Eq. (4) since $\kappa = 2$ when $n = 2$.

Figure 2e, f shows the resulting values of $a(h_c)$ for the probe used in the current work. The prediction of the contact radius is not reliable for $h_c < 5$ nm because the variability in the experimental results is too great in this region, but the near-contact geometry is shown as a power law cap as a dashed line in Fig. 2f. The geometric parameters of the tip at different indentation depths are reported in Table 1. A more comprehensive table and the tip contact radius function example can be found in Supplementary Table S1 and Supplementary Fig. S2.

Finally, a best-fit polynomial from $h_c = 10$ to 200 nm is derived for $a(h_c)$:

$$a = d_0 + d_1 h_c + d_2 h_c^2 + d_3 h_c^3 + d_4 h_c^4 + d_5 h_c^5 + d_6 h_c^6 \tag{10}$$

This typically covers the complete range of $h_c$ values obtained from the corneocyte indentation measurements. The first two terms in Eq. (10) have physical relevance as representing a perfect truncated cone, with the higher terms accounting for deviations from this profile.

To validate this calibration method, a sample of polymethyl methacrylate (PMMA) sheet (Merck KGaA, Darmstadt, Germany) was indented with two different AFM probes and with different force setpoints. The values of elastic modulus obtained from the indentation data were satisfactorily close to those in the literature (see Supplementary Material S6).

## Nanoindentation and stress relaxation of corneocytes

The measurements were done under controlled ambient conditions of 25.5 °C and 35% relative humidity (RH), which corresponds to a nominal dry state for these cells, comparable to the conditions on the skin surface in their native state. The loading-unloading data at a speed of 0.5 μm/s obtained for cells attached to tape exhibited a hysteresis loop (Fig. 3a), which was much more significant than observed for the PDMS (Fig. 2b). This may be the result of viscous, plastic and/or adhesive components[7,23–26]. For both elastoplastic and viscoelastic deformations, after the removal of a load, a residual imprint may remain on the surface[27]. For elastoplastic materials, this imprint is irreversible, while for viscoelastic deformations, the imprint is time-dependent and may recover given sufficient time. In the current experiments, a permanent indent was observed on the surface by imaging the cells after indentation (Fig. 3c–e).

Although the corneocytes exhibited elastoplastic behaviour, the unloading data for such materials are purely elastic and, consequently, the Young's modulus is given by the following expression[28]:

$$E = \frac{\sqrt{\pi}}{2}(1 - v^2)\frac{S_0}{\sqrt{A_{max}}} \tag{11}$$

where $S_0$ is the stiffness at the maximum force of the unloading curve, $F_{max}$, corresponding to the maximum indentation depth, $h_{max}$, and $A_{max} = \pi a^2$ is the contact area[24]. The Poisson's ratio for corneocytes, $v$, was assumed to be 0.4, considering what has been described for keratin[29]. The tip radius function obtained from PDMS nanoindentation (Eq. (10) and Supplementary Fig. S2) was used to obtain the contact radius at $h_{max}$. For an elastoplastic contact, the contact depth, $h_c$, is related to $h_{max}$ by the expression[28]:

$$h_c = h_{max} - h_s \tag{12}$$

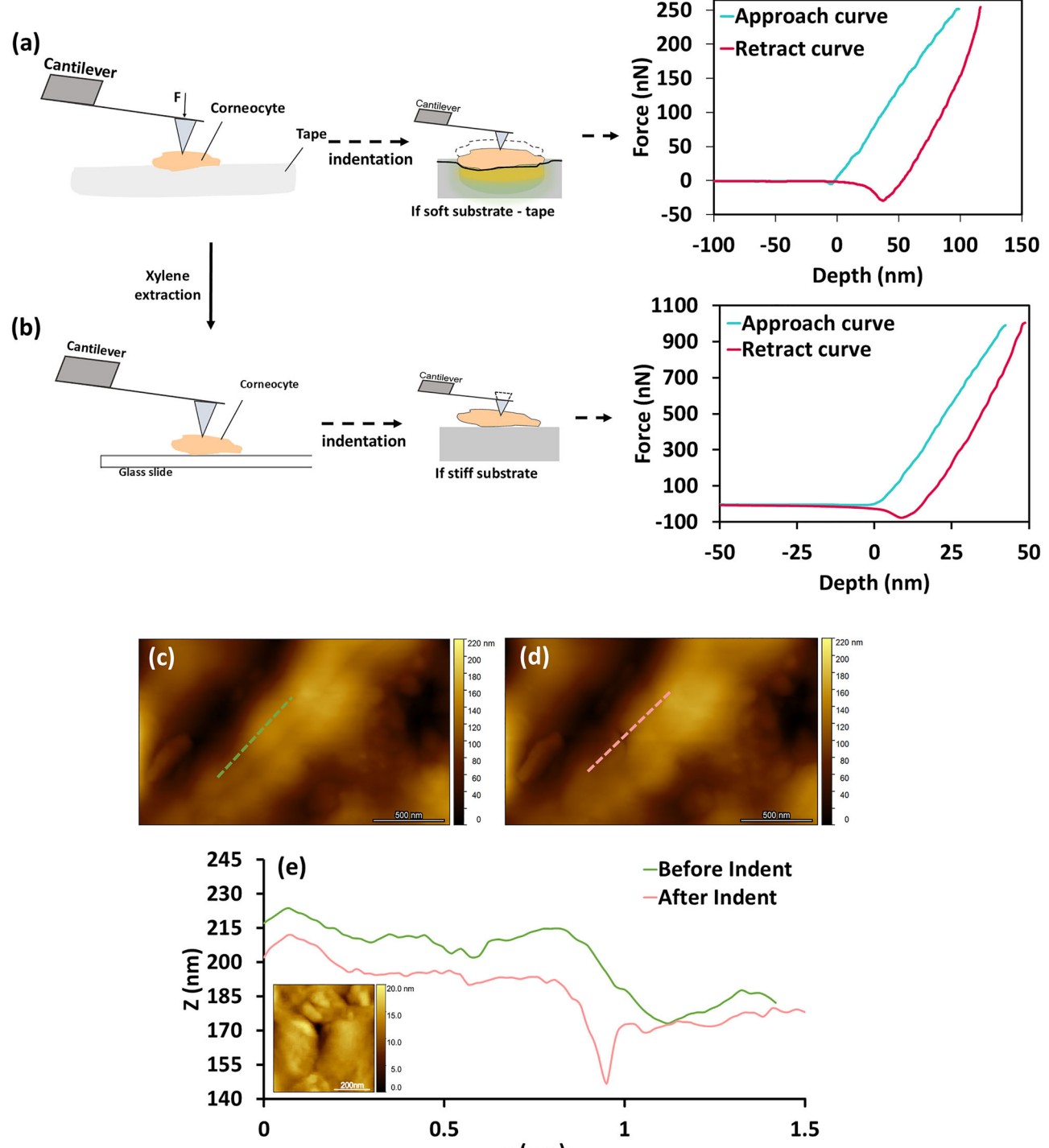

**Fig. 3 | AFM nanoindentation of corneocytes. a** Schematic steps used to measure the mechanical properties of corneocytes that illustrate the effect of the compliance of the supporting substrate. The cells were initially adhered to Sellotape (soft substrate) and **b** were then extracted to a glass slide (stiff substrate) using xylene to dissolve the adhesive. **c** and **d** correspond to the surface topography of a corneocyte before and after indentation of cells on a glass slide with a force of 2 μN. Scale bar = 500 nm. **e** The line profiles from (**c**) and (**d**) with an inset showing a zoomed topographical AFM image from (**d**) of the permanent indent remaining on the surface of the cell. Inset scale bar = 200 nm.

where $h_s$ is the surface elastic deflection at the perimeter of the contact (Fig. 4b), which is given by the following expression[28]:

$$h_s = \phi \frac{F_{max}}{S_0} \tag{13}$$

The initial unloading stiffness, $S_0 = dF/dh$ was obtained by differentiating the following polynomial fit to the upper 80% of the unloading curve and determining the slope at $h_{max}$[23] (Fig. 4a):

$$F = C_e h^2 - \left(2C_e h_f\right)h + C_e h_f^2 \tag{14}$$

where $C_e$ and $h_f$ are fitting parameters. The geometric factor, $\phi$, was assumed to be 0.73 for a conical indenter, following the work of Oliver and Pharr[30] and considering that the load index, $m$, of the unloading curves of corneocytes was $m \approx 2$. Equation (14) is an expansion of the following power law with $m = 2$:

$$F = C_e(h - h_f)^m \tag{15}$$

The overall values of Young's moduli of the volar forearm corneocytes on the tape used to harvest the cells and on a microscope glass slide are presented in Table 2. Cells adhering to tape showed smaller values in the range of 12–66 MPa, compared to the most recent literature that reports values in the range of 250–500 MPa[7,8]. However, cells on a microscope glass slide had much greater values ranging from 200–1562 MPa (Table 2 and Fig. 4c–h), which reflects the advantage of a stiff supporting substrate. Considering the significant effect of the soft substrate (tape) on the nanoindentation results, the values of Young's modulus obtained for the cells attached to the tape should not be compared with those of cells attached to the glass slide. In fact, as previously studied for fibroblasts on soft gels, the effect of substrate deformability on cell stiffness measurements can be considerable[15]. The adhesives that constitute regular tapes, such as the one used in the current study, belong to the family of polyacrylates, which are characterised by relatively small values of Young's modulus ($\sim 2 \times 10^4$–$5 \times 10^5$ Pa)[31], when compared to comparatively stiff SC in a relatively dry state[32]. Therefore, the effect of the deformability of the adhesive cannot be excluded for indentations of cells attached to a tape. Due to the complex AFM tip geometry, the wide variation of the thickness of the adhesive, and the mechanical behaviour of the corneocytes described in the current study, it was not possible to correct this effect using, for example, the CoCS model[15]. The trends in the Young's moduli corresponding to the two substrates for the three participants are not systematic, is a result of a significant variation in the thickness and surface topography of the adhesive (Supplementary Fig. S8).

Although greater than reported previously, the Young's moduli for corneocytes on glass slides are consistent with those obtained for SC, of about 2 GPa at 30% RH, and decreasing with increasing RH ($\sim 1$ GPa at 50% RH)[32]. Since corneocytes in this study are in a relatively dry state (35% RH), they display mean values that are similar to organic polymers in the glassy state[33,34], which has also been observed for keratin, the major component of these cells[35]. The interpretation of the variability found between cells and participants was not the subject of the current study. Nonetheless, it may be due to the limited number of cells and participants or dependent on the maturation level of individual corneocytes[11].

Such variation is not uncommon when performing mechanical characterisation using AFM, particularly for biological samples. For example, previous studies have reported a large range of values when indenting hair[36,37]. Biological materials are not homogeneous, and, in the case of corneocytes, the specific local arrangement of the keratin network and intracellular junctions, differences in the architecture of the CE or other unaccounted factors, such as the surface topography, may give rise to such distributions.

It is well known (see Supplementary Material S8) that there is a tendency for lateral deflection of an AFM probe during indentation that is intrinsic to the elastic deflection of a tilted cantilever beam with a finite tip length (see Fig. 3). For the cantilever used in the current work, the lateral tip end deflection is estimated to be 37% of the normal deflection (see Supplementary Material S8). In practice, this percentage will be reduced by tangential contact forces that cause parasitic reductions in cantilever end deflection angles. This parasitic bending at the end of the cantilever will result in apparent reductions of the normal force and, hence, to underestimates in derived material property values. However, since the values of the elastic modulus obtained for PMMA are similar to those quoted in the literature (see Supplementary Material S6), it is unlikely that this leads to major systematic errors in the values given in Table 2 for cells attached to glass slides.

Typical force-relaxation data are shown in Fig. 5. The aim was to determine the parameters for a viscoplastic material model that would be more representative of the large-strain mechanical properties of corneocytes. The data were analysed using a Prony series for which two relaxation times were sufficient (Fig. 5c)[22]:

$$F(t) = B_0 + B_1 e^{-t/\tau_1} + B_2 e^{-t/\tau_2} \tag{16}$$

where $B_0$, $B_1$, and $B_2$ are fitted force-relaxation coefficients, $\tau_1$ and $\tau_2$ are fitted relaxation times, and $t$ is the time.

The corneocytes deform plastically and may be characterised by the hardness, $H$:

$$F = \pi a^2 H \tag{17}$$

Moreover, the mean contact stress, $\bar{\sigma}$, is equal to the hardness:

$$\bar{\sigma} = H = \frac{F_{max}}{A_{max}} = \psi \sigma_Y \tag{18}$$

where $\sigma_Y$ is the uniaxial yield stress and $\psi \approx 3$ is the constraint factor assuming that the material has fully yielded[38].

For materials that show viscoplastic behaviour, the hardness $H$ is strain rate dependent. Following the approach described by Yan et al.[22], the time-invariant parameter $H$ in Eq. (18) may be replaced by the time-dependent value $H(t)$ to describe transient measurements, where $H(t)$ can then also be written as a Prony series:

$$H(t) = C_0 + C_1 e^{-t/\tau_1} + C_2 e^{-t/\tau_2} \tag{19}$$

The hardness relaxation coefficients $C_0, C_1, C_2$ are related to $B_0, B_1, B_2$ (Eq. (16)) as suggested in Refs. 22,39 by the following expression:

$$C_i = \frac{B_i}{\pi a^2} \tag{20}$$

where $i = (0, 1, 2)$ and $a$ is obtained from the analysis of the unloading data as described previously. The instantaneous hardness, $H_0$, and the long-term hardness, $H_\infty$, can be estimated using the following expressions:

$$H_0 = \sum_{i=0}^{2} C_i \tag{21}$$

and

$$H_\infty = C_0 \tag{22}$$

The force-relaxation data were first fitted to Eq. (16) from which the coefficients $B_0$, $B_1$, $B_2$ were obtained. Then the coefficients $C_0$, $C_1$, $C_2$ were calculated using Eq. (20). The derived material parameters describing the viscoplastic behaviour of the corneocytes are presented in Table 3.

## Discussion

AFM has emerged as the gold standard for studying the mechanical properties of biological cells. Although it was not developed to perform nanoindentation experiments, the nano-resolution allows not only the characterisation of soft samples, but also the ability to generate very shallow indentations in thin samples. Furthermore, the possibility of combining topographical and mechanical analysis considerably enhances the potential of this technique. AFM nanoindentation typically involves obtaining loading-unloading cycles, in which a maximum loading force setpoint is selected by the user, with subsequent retraction of the tip. Although loading curves contain detailed information about the forces experienced by the AFM probe, the type of deformation (elastic or inelastic) cannot be determined solely by these data and must be elucidated from the unloading

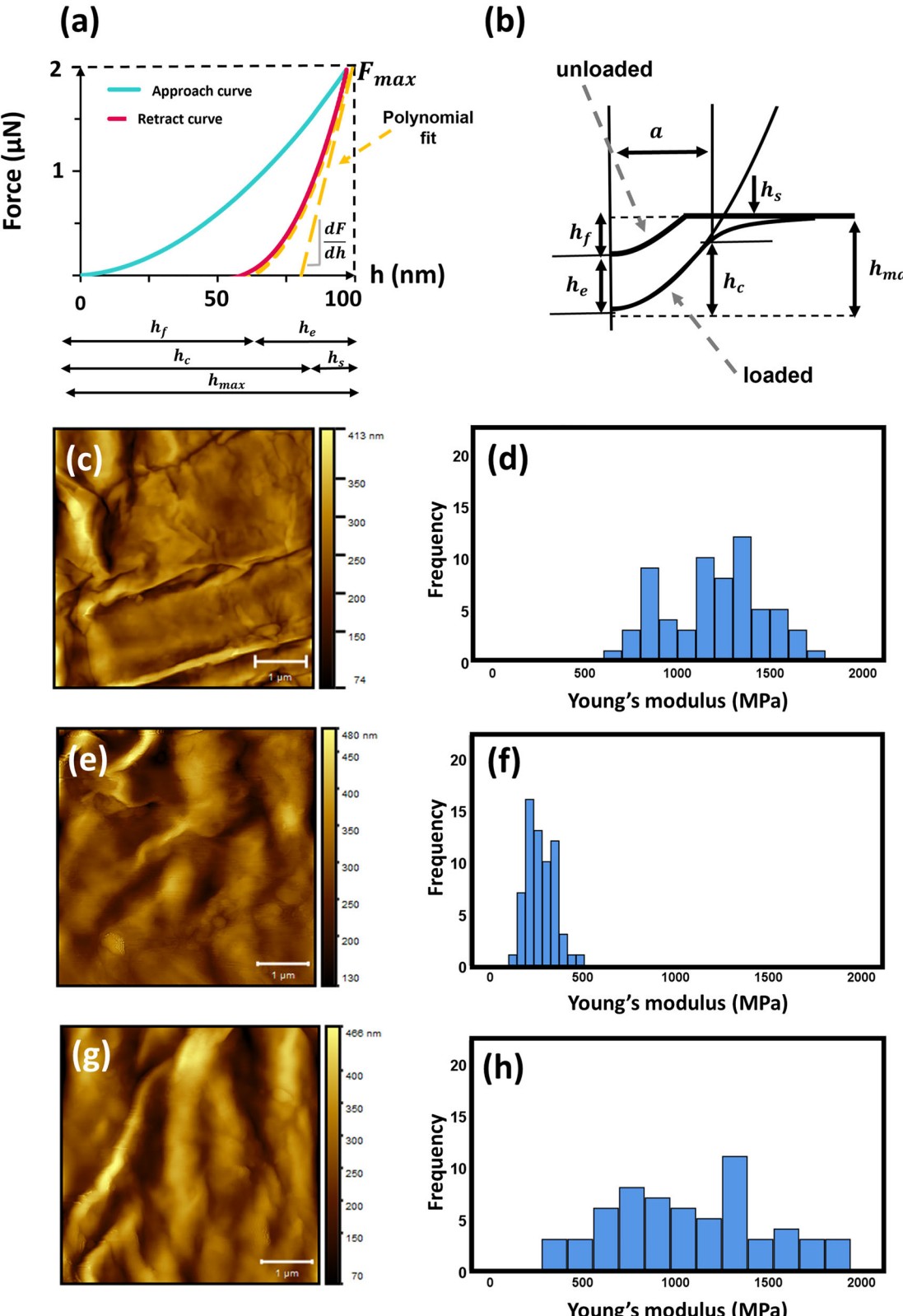

**Fig. 4 | AFM force curve and material deformation schematic diagrams, Young's moduli histograms and corresponding images for three corneocytes. a** Graphical representation of the load as a function of the indenter displacement showing the quantities used in the analysis and **b** a schematic interpretation of these quantities, adapted from Ref. 23. $a$ is the contact radius; $h_{max}$ is the maximum indentation depth; $h_c$ is the contact depth; $h_s$ is the surface elastic deflection at the perimeter of the contact; $h_f$ is the final depth of the residual hardness impression; $h_e$ is the elastic component of the displacement. TM-AFM images of corneocytes and histograms showing values of Young's modulus calculated from 64 force curves in that region for **c** and **d** Cell 1 of P1, **e** and **f** Cell 3 of P2 and **g** and **h** Cell 4 of P3. The cells were fixed on a glass slide, and the maximum force setpoint was 2 µN. Scale bars = 1 µm.

curves, during which elastic recovery occurs as the residual indentation impression is formed[28]. However, a viscous component might still exist during unloading. This can be reduced by decreasing the speed of indentation and by allowing a period of relaxation, which here was 4 s. The adhesion component, which can be assessed by the negative part of the unloading curves, was ignored in this study since it was largely attenuated using the aqueous SDS solution for PDMS and was small compared to the positive part of the load curves for the corneocytes. Consequently, loading-pause-unloading cycles were performed on individual corneocytes (Fig. 5) with the Young's modulus values obtained from the unloading curves, since hysteresis was observed as a result of inelastic deformation (Fig. 3).

Corneocytes are flat dead cells (without a nucleus or organelles) with typical thicknesses and widths of 0.2–1.0 μm and 10–30 μm, respectively[11,40]. They are characterised by a rigid cross-linked protein envelope that is termed the cornified envelope (CE), to which a lipid envelope is covalently attached (CLE)[11,41]. They encompass an internal matrix that is mainly

### Table 2 | The Young's moduli of volar forearm corneocytes, either attached to tape strips or fixed to glass microscope slides

| Participant ID | Young's modulus (MPa) | |
|---|---|---|
| | Tape | Glass slide |
| 1 | 15.9 (12.5–21.5) | 1183 (482–1263) |
| 2 | 21.6 (13.8–66.6) | 424 (200–533) |
| 3 | 31.3 (16.3–39.5) | 1105 (423–1562) |

The maximum force setpoint used was 250 nN for the tape substrate and 2 μN for the glass substrate. The results for three participants are presented as the median and range (for 5 cells per participant and 64 force curves per cell).

composed of keratin filaments and natural moisturising factors (NMF)[42]. The thickness of the cornified envelope varies from 20 to 50 nm[8]. The forces required to indent corneocytes were relatively high (1–2 μN) compared to many other cell types (usually a few nN)[43]. In fact, corneocytes cannot be compared to typical cultured cells since their relatively rigid protein envelope surrounds the stiff polymer keratin[11]. At low relative humidities (RHs), the SC exhibits glassy state values of the Young's modulus, in the range 1–2 GPa[32], and it is reasonable to assume that this arises mainly from keratin, which was found to account for 85% of the total protein content of the SC[44]. Previous studies of corneocytes with AFM have measured values of the Young's modulus in the range 250–500 MPa[7,8]. Such values are an underestimation on the basis of the various uncertainties of the AFM protocols used, particularly that of the selection of the substrate. Here, the mean Young's modulus of corneocytes on a stiff substrate (microscope glass slide) was in the range 0.2–1.6 GPa (Table 2), for an RH of about 35%, which is consistent with the early works of Park and Baddiel on SC[32]. This is much greater than that calculated for those adhered to a soft substrate by a factor of ~20. Unfortunately, the curves obtained on tape could not be simply corrected using the recently developed CoCS model[15], because of the complex elastoviscoplastic behaviour of corneocytes. The current results are also consistent with those recently published by Boonpuek et al.[45], who presented values of Young's modulus in the range 0.6–1.0 GPa for single corneocytes attached to an aluminium plate. The results are based on the JKR model for the adhesive surface energy of elastic solid-solid contact between two spheres.

A number of attempts have been employed to study the properties of the different components of corneocytes (CLE, CE and keratin matrix), by the use of nanoneedles[7] or using a tomography method relying on the Hertz model[8]. However, they involve the uncertainties in the protocols identified in the current work. Corneocytes may be treated as a composite of two layers: a keratin matrix and a CE film. Independent of which component

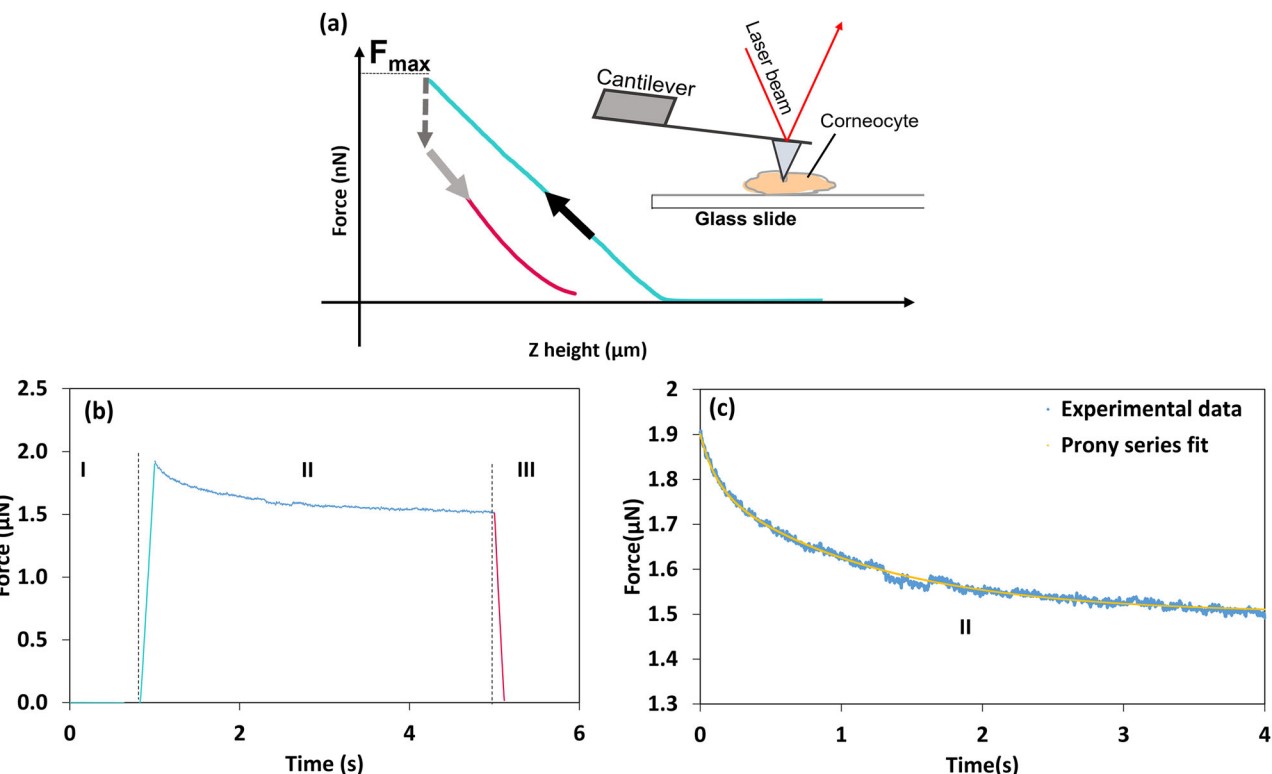

**Fig. 5 | AFM force-relaxation schematic diagram and typical data for corneocytes. a** Schematic diagram of the stress-relaxation measurement consisting of an approach curve, a pause segment, and a retraction curve. The measurements were performed on cells fixed on a glass slide. **b** Representative force as a function of time data consisting of three stages: (I) loading curve up to a maximum force setpoint of 2 μN, followed by (II) a 4 s pause at a constant vertical AFM head height (Z height) and, finally, (III) the retraction of the probe from the cell surface. **c** The force-relaxation curve was fitted by a Prony series to obtain the time-dependent mechanical properties of corneocytes.

**Table 3 | Material parameters of the volar forearm corneocytes calculated from force-relaxation AFM curves**

| Participant ID | Relaxation times | | Hardness | |
|---|---|---|---|---|
| | $\tau_1$ (s) | $\tau_2$ (s) | $H_0$ (MPa) | $H_\infty$ (MPa) |
| 1 | 0.12 (0.10–0.17) | 1.99 (1.34–2.17) | 198 (128–200) | 151 (90–164) |
| 2 | 0.16 (0.15–0.19) | 2.11 (2.07–5.46) | 127 (47–162) | 76 (21–130) |
| 3 | 0.14 (0.11–0.16) | 2.40 (1.57–4.22) | 159 (46–188) | 92 (13–128) |

The results are presented as median and range (for 5 cells per participant and 64 force curves per cell).

gives the mechanical strength to the cell, to "quantify" their individual contributions might not be possible because:

1. If the CE is softer than the keratin matrix: considering the restriction for nanoindentation of thin films (< 10% of sample thickness, see Supplementary Material S9)[46] and the thickness of the CE to be of about 50 nm, indentations would need to be limited to only ≤5 nm.
2. If the CE is stiffer than the keratin matrix, the keratin matrix would be compressed in an analogous way to the tape presented in this work, so that the stiffness would mostly arise from the matrix.

That is not to say that differences between cells at different depths[8] or different people[7] do not result from the different components of the cells, but to reveal which component is responsible for such differences might be difficult to deconvolute by nanoindentation of cells in the dry state. While there have been models developed for layered elastic and elastoplastic solids, it is complicated analytically to obtain closed-form solutions, and generally it is necessary to apply numerical methods such as finite element analysis[47]. Existing analytical models have not been developed for elastoviscoplastic solids or power law indenters, and, consequently, a numerical method would be required to quantify the mechanical properties of the two layers.

The current work has established that corneocytes are elastoplastic with a time-dependent yield stress (Table 3, Fig. 5). Material models relate the stress to the strain or strain rate. To obtain a rate-dependent material model from the time-dependent data, it may be assumed that the relaxation arises from a conversion of the elastic, $\varepsilon_e$, to the plastic, $\varepsilon_p$, strain so that the total strain, $\varepsilon$, may be decomposed additively as follows:

$$\varepsilon = \varepsilon_e + \varepsilon_p \tag{23}$$

Assuming $d\varepsilon/dt = 0$ and a constant value of the Young's modulus $E = \sigma/\varepsilon_e$, where $\sigma$ is the unconfined bulk stress:

$$d\varepsilon_p/dt = -d\varepsilon_e/dt = -d(\sigma/E)/dt \approx -\frac{1}{E}d\sigma/dt \tag{24}$$

Substituting the derivative of Eq. (19) with the mean stress, $H = \bar{\sigma} = 3\sigma$, the uniaxial strain rate is given by:

$$d\varepsilon_p/dt = \frac{3}{E}\left(\frac{C_1}{\tau_1}e^{-t/\tau_1} + \frac{C_2}{\tau_2}e^{-t/\tau_2}\right) \tag{25}$$

so that a material model can be obtained from a plot of $\sigma$ as a function of $d\varepsilon_p/dt$. Generally, viscoplastic materials are described by a stress overshoot model to account for the strain rate hardening[48]. However, the Herschel-Bulkley model is applied here since it is equivalent for uniaxial deformation[49]:

$$\sigma = \sigma_Y + k\left(d\varepsilon_p/dt\right)^j \tag{26}$$

where $\sigma_Y$ is the quasi-static uniaxial yield stress, $k$ is the plastic flow consistency, and $j$ is the plastic flow index. It should be noted that this is a rigid-viscoplastic model since it is assumed that yielding occurs at a zero strain. That is, elastic deformation within the yield surface is treated independently.

Example fits of Eq. (26) to the experimental data for the three participants are shown in Fig. 6 together with histograms of the yield stresses that reflect the variation in the values of the Young's modulus described earlier. The median values of the material parameters are given in Table 4. They are all of similar order, but further studies are required to elucidate differences between different anatomical locations and under different conditions (moisture content, liquid immersion, etc.).

In summary, the current paper presents nanoindentation protocols to measure the mechanical properties of corneocytes, addressing the most important uncertainties of using AFM to describe the mechanical properties of SC cells. It was concluded that the tape used for stripping corneocytes is unsuitable as a substrate for AFM indentation measurements since it is much softer than the cells and hence leads to erroneously small Young's modulus values. Moreover, the nominal stiffnesses of the cantilevers and geometries of the tips stated by the supplier should be taken with caution, particularly when using standard TM cantilevers for nanoindentation of relatively stiff materials. The use of PDMS as a reference material provides an accurate, effective geometry of a tip, which demonstrates that the often-employed Hertz analysis for a parabolic geometry is unsuitable. Furthermore, at small stresses, corneocytes exhibit elastic behaviour, but visco-plastic behaviour is observed when the stress is equal to the yield value, which can be described by the Herschel-Bulkley material model. The current methodology has been used to study the effects of water activity on the mechanical properties of corneocytes[50] and to describe the mechanical behaviour of corneocytes collected from several anatomical sites[51].

## Methods
### Corneocyte collection
Corneocytes were collected following informed consent from three healthy adult participants (two males and one female), aged 26–29 years. All ethical regulations relevant to human research participants were followed in accordance with the UK regulations and the Declaration of Helsinki. Ethics approval was obtained from the University of Birmingham Research Ethics Committee – ERN-19-1398A. Samples were collected using the tape stripping method (Sellotape, UK). A first layer of corneocytes from the volar forearm was removed and discarded to avoid the presence of contaminants, such as clothing fibres. The second tape strip was used for the AFM measurements. Half of the tape was used directly, while the other half was pressed on a glass slide and immersed in xylene overnight. This detaches the tape by dissolving the tape adhesive, leaving the corneocytes transferred to the glass. Attached cells were further washed (2× for 30 min) in xylene. FTIR spectra of corneocytes before and after xylene extraction did not reveal any structural differences (Supplementary Material S4). Topographical images of the cells (40 × 40 μm) and a zoom-in TM image (5 × 5 μm) from the central region were obtained (workflow described in Supplementary Material S2).

### Environmental Scanning Electron Microscopy (ESEM) and Tapping Mode (TM) AFM for determining the probe geometry
The overall geometry of the AFM tip was measured using ESEM. The sample was sputter-coated with platinum and imaged using a Philips XL30 FEG ESEM operating in high vacuum mode with an accelerating voltage of 20 kV. ImageJ® version 1.53a (National Institutes of Health, Bethesda, MD, USA) was used to process the ESEM images. The effective tip radius was also evaluated by imaging a calibration sample (TipCheck, Apex Probes, UK) in

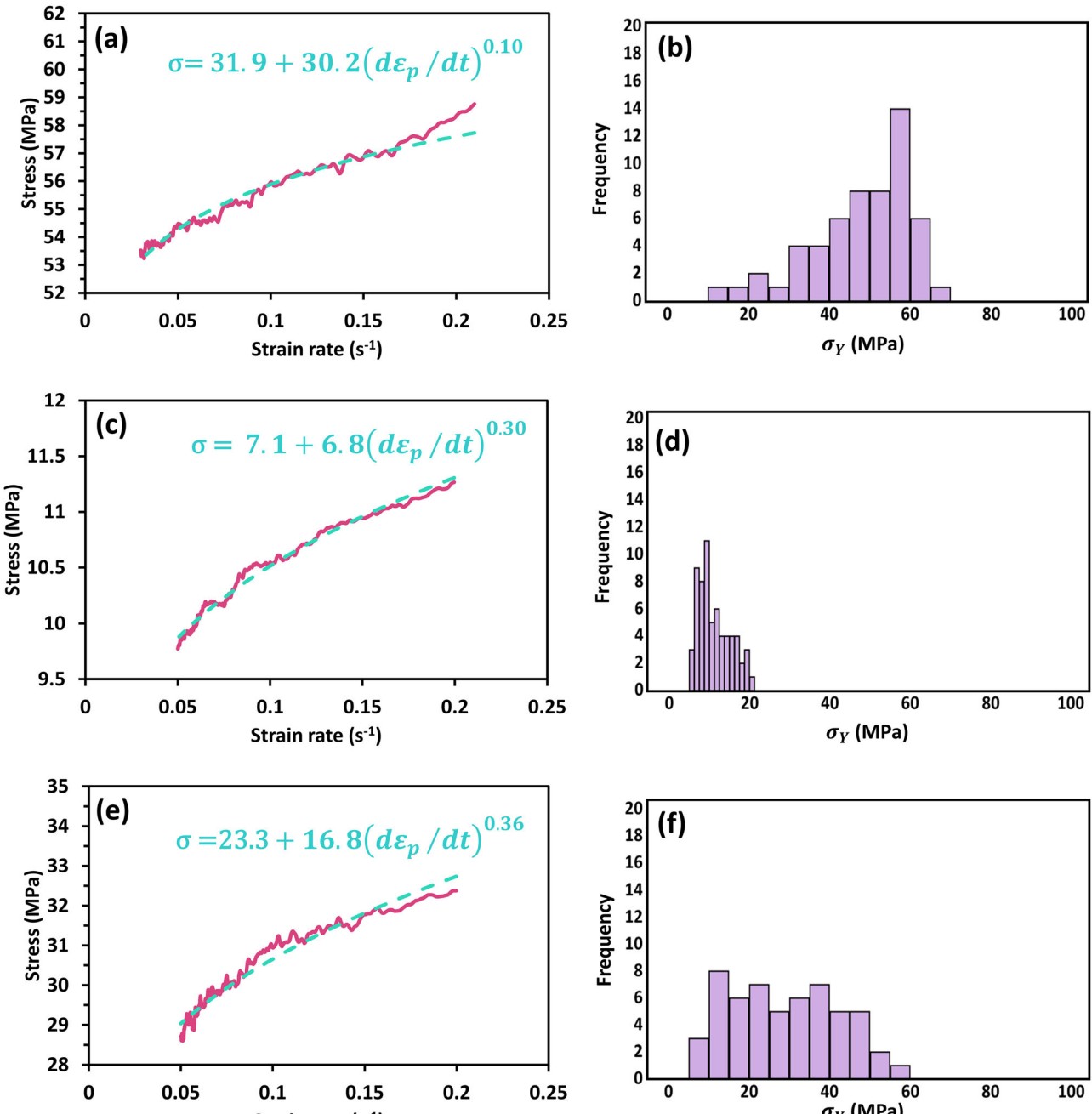

**Fig. 6 | AFM viscoplastic data curve fits and yield stress histograms for corneocytes.** Typical Herschel-Bulkley material model fits from single relaxation curves using AFM for (**a**) Cell 2 of P1, **c** Cell 3 of P2 and (**e**) Cell 4 of P3 and corresponding histograms (**b**), (**d**) and (**f**) of the calculated yield stresses based on 40–50 stress-relaxation curves. The solid lines are the experimental data. The dashed lines correspond to the Herschel-Bulkley viscoplastic fits.

Tapping Mode (TM) and using a blind tip estimation (partial) algorithm included in Gwyddion, which is an open-source software package for AFM image processing[19]. This algorithm iterated over all the surface data and refined the tip point according to the steepest slope in the direction between the tip point and the tip apex.

**Silicone elastomer (PDMS) sample preparation for indenter geometry calibration**
Polydimethylsiloxane (PDMS) elastomer was selected as an ideal material for calibrating the geometry of the indenters since it has a relatively small Young's modulus that is appropriate for the cantilever stiffnesses used for the corneocytes, and it is homogeneous, isotropic and, at small strains and strain rates, it is linear elastic. It was cured as in Wang et al.[52]. While there

have been a number of studies suggesting that the Young's modulus of PDMS increases with decreasing indentation depths, recently it has been shown that this arises from errors in the surface detection[53]. In the current work, the PDMS was prepared by mixing Sylgard 184 silicone elastomer base and Sylgard 184 silicone elastomer curing agent (Merck KGaA, Darmstadt, Germany) in a mass ratio of 5:1 (base/curing agent). A PDMS sheet (5 mm in thickness) was made by pouring the mixture into a flat-bottom polystyrene dish, and the PDMS was degassed under vacuum to remove air bubbles. It was cured in an oven at 65 °C for 1 h. The sheet was cut with a biopsy punch, and the cast side facing air was used for force spectroscopy measurements. This reference specimen had a mean roughness, $S_a$, calculated from AFM Tapping Mode of 2.1 nm (from a 25 μm$^2$ AFM image – see Fig. 2a).

**Table 4 | Herschel-Bulkley material parameters for volar forearm corneocytes**

| Participant ID | $\sigma_Y$ (MPa) | $j$ | $k$ (MPa s$^j$) | $R^2$ (mean ± 1 SD) |
|---|---|---|---|---|
| 1 | 48 (29–53) | 0.33 (0.28–0.36) | 16 (15–23) | 0.971 ± 0.003 |
| 2 | 21 (7–41) | 0.34 (0.14–0.43) | 14 (10–22) | 0.971 ± 0.009 |
| 3 | 29 (4–43) | 0.28 (0.19–0.42) | 25 (11–42) | 0.955 ± 0.030 |

The material parameter results are presented as the median and range (for 5 cells per participant and 64 force curves per cell).

## AFM experiments

AFM imaging and indentation experiments were performed in air, operated in tapping mode (TM) and Force Spectroscopy mode, respectively (Nanowizard 4, Bruker, JPK BioAFM, Berlin, Germany). Indentation involved a standard silicon TM tip (NCHV-A, Bruker AFM Probes, Inc). On the basis of the values of Young's modulus described in the literature for corneocytes (hundreds of MPa), AFM cantilevers with a nominal spring constant of 40 N/m (relatively stiff) and a nominal resonance frequency of 320 kHz were selected, with the advantage of being able to perform a topographical characterisation using TM-AFM of the regions of interest. All measurements were conducted under a controlled temperature and humidity of 25.5 °C and 35% RH. The loading and unloading data were analysed using customised Matlab scripts (MathWorks Inc).

Force Spectroscopy mode involves the nanoindentation of a sample by a tip located on a cantilever. Raw cantilever deflection data are recorded as a change in amplitude (in Volts), which can be converted to force by calibrating both the deflection sensitivity of the system and the spring constant of the cantilever. The deflection sensitivity is calculated before or after each set of experiments by pressing the tip onto a relatively rigid material (e.g., glass microscope slide or sapphire, depending on the cantilever stiffness). Assuming there should not be a significant indentation on a stiff substrate, the deflection of the cantilever represents the sensitivity of the system (nm/V). Here, the sensitivity of the equipment was calibrated after each set of experiments by pressing onto a sample of sapphire, considering that the probe is made of etched silicon, which has a Young's modulus of about 160 GPa[54] and, thus, is greater than that of glass ($E = 66$ GPa[54]), but much smaller than that of sapphire ($E = 470$ GPa[54]). As shown in Supplementary Material S3, this resulted in a smaller sensitivity when pressing sapphire (18–22 nm/V) than when pressing on a microscope glass slide (28–35 nm/V).

The spring constant or stiffness of the cantilever (N/m) was calibrated using the Sader method[55], which considers the size of the cantilever, as well as a thermal tuning spectrum as a result of the thermal motion, to which a single harmonic oscillator is fitted in order to extract the resonance frequency and quality factor. In the current study, the cantilever had a spring constant of 28.5 N/m, which, although being in the range of values given by the supplier (20–75 N/m), is significantly different from the nominal value of 40 N/m. Consequently, the spring constant of cantilevers should be calibrated to ensure corrected values of force.

## Nanoindentation of PDMS

Nanoindentation measurements were performed on the PDMS specimen in a solution of 0.1% w/w aqueous sodium dodecyl sulphate (SDS) to attenuate adhesion forces[56]. Three different regions were evaluated with 64 force curves (in each 5 × 5 µm region) at a maximum setpoint of 250 nN. The loading and unloading force data as a function of tip displacement were obtained using the AFM Force Spectroscopy mode, from which the geometry of the probe was determined as summarised above and described in more detail in Supplementary Material S1.

## Nanoindentation of corneocytes

Nanoindentation measurements were carried out on corneocytes adhered to Sellotape (Fig. 3a) and after being extracted with xylene and fixed to a microscope glass slide (Fig. 3b). The selection of the substrate is very important since the tape used here as a substrate is composed of two main components: a backing film and an adhesive, to which the corneocytes adhere. These adhesives usually belong to the family of polyacrylates, which are characterised by a relatively small value of Young's modulus of less than tens of MPa[31]. Nanoindentation should be performed to a depth of <10% of the sample thickness in order to avoid artefacts arising from the substrate[46,57] (see Supplementary Material S9). However, this is only applicable for stiff substrates. When the substrate is softer than the sample, as in the case of tape, even for techniques at the nano length-scale, such measurements will result in the sample being pushed onto the substrate as it is indented (Fig. 3a).

For a total of 5 cells per subject, 64 loading and unloading force curves were collected in a 5 × 5 µm region at a velocity of 0.5 µm/s. This included a dwell time of 4 s after loading to allow any viscous component to relax. For each corneocyte, the maximum force setpoint was adjusted by ensuring that the maximum indentation depth was not greater than 10% of the cell thickness (≤1 µm) in order to avoid artefacts arising from the supporting substrate[46,57] (see Supplementary Material S9). The maximum applied force was 250 nN for cells adhered to Sellotape and 2 µN for cells fixed on a microscope glass slide in order to obtain similar indentation depths with both methods ranging between 30 and 100 nm (Fig. 3). Force-relaxation measurements of the corneocytes were obtained during the dwell time of 4 s at constant vertical AFM head height, after the trigger threshold was reached (Fig. 5). Individual force-time curves (64 curves in a 5 × 5 µm matrix) were obtained for five corneocytes per sample, at a loading velocity of 0.5 µm/s.

## Statistics and reproducibility

Unless stated otherwise, all derived material property parameter values are quoted as mean ± one standard deviation. Biological replicates consisted of corneocytes from three healthy adult participants, with five cells per participant being measured. Technical replicates included 64 force curves measured per corneocyte. Steps taken to ensure reproducibility included maintaining controlled environmental conditions and calibrating the shape of each tip using a reference specimen of PDMS. Accuracy was demonstrated by evaluating the mechanical properties of a reference material, PMMA.

## Reporting summary

Further information on research design is available in the Nature Portfolio Reporting Summary linked to this article.

## Data availability

Source data for all plots presented in this manuscript can be found as Supplementary Datasets 1–4. Raw data is available from the corresponding author upon request.

## Code availability

The MATLAB scripts used in this study are available on GitHub (https://github.com/anaevora/AFM-corneocyte.git) and have also been uploaded as Supplementary Software 1. The code is released under the MIT license. MATLAB R2022a was used. The scripts can be accessed without restrictions.

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

## Acknowledgements
This work was supported by the European Union's Horizon 2020 research and innovation programme under the Marie Skłodowska-Curie grant agreement No. 811965 (Project STINTS - Skin Tissue Integrity under Shear).

## Author contributions
A.E. conducted all the experiments and wrote the manuscript with the assistance of the other authors. M.A., S.J. and A.E. developed tip geometry analysis. Z.-H.Z., Z.Z. and M.A. developed the viscoplastic model. All authors revised the data analysis and edited the manuscript.

## Competing interests
The authors declare no competing interests.
