## [Transparent Peer Review file · Communications Biology]

Atomic force microscope protocols for characterising the elastoviscoplastic biomechanical properties of corneocytes

Corresponding Author: Professor Michael Adams

Version 0:

Reviewer comments:

Reviewer #1

(Remarks to the Author)

The paper "Atomic force microscope protocols for characterizing the elastoviscoplastic biomechanical properties of corneocytes" by Evora et al. addresses very important procedural considerations for cell stiffness measurements. The authors correctly point out that there is considerable variability in data obtained and, although these measurements have been in use for over two decades, there is room for improvement in both data acquisition and analysis approaches. Cell stiffness measurements provide a wealth of information with often significant implications not only in the lab but also in a clinical setting. Such measurements have the potential to diagnose and predict disease outcomes in addition to, of course, furthering our understanding of vital biological processes. Thus, it is important that we get this right and are consistent in our approaches. The authors address tip geometry, substrates and modelling.

An initial concern when looking at the data was the use of relatively high indentation forces. In past studies, we have found that using indentation forces in excess of 1 nN could damage living cells and result in indentations that exceeded 1 micron (less than 1 micron ensured good fit with the Hertz model). However, the authors study focuses on corneocytes (anuclear, dead cells) that are unlikely to be damaged by higher indentation forces. The authors appropriately address this point in the discussion.

The manuscript offers a thorough investigation into the tip geometry, a very important aspect that must be taken into consideration when selecting the appropriate model to fit cell stiffness data. The authors conclude that corneocytes exhibit viscoplastic behavior and employ the Herschel-Bulkley material model to fit their data.

In summary, this is a thorough investigation into an important approach that needs more consistency. I recommend publication with no revisions.

Reviewer #2

(Remarks to the Author)

This is a potentially very interesting and important attempt to introduce some standards in the measurements of corneocytes. However, multiple technical issues prevent this reviewer from recommending this work, at least in its present form. Just a few major technical issues:

1. The use of sharp indenter. "wide range of Young's modulus values reported in the literature.." Since it is a new area, it is plausible to expect that many measurements might be simply incorrect. For example, there are quite a lot of works published demonstrating that the use of excessively sharp probes cannot be utilized to measure the mechanical properties of cells [P. H. Wu, et al, Nature Methods, 2018, 15, 491] and polymers [M. E. Dokukin et al, Macromolecules, 2012, 45, 4277-4288.]. So it is not clear why the author didn't use quite generally accepted these days spherical indenter as with well-defined geometry.
2. The use of a reference material to restore the probe geometry. "a silicone elastomer reference material for determining the effective indenter geometry". The method which was recognized about 10 years ago as putting "dust onto the carpet". When using a sharp indenter, the recovered geometry of the indenter will depend on the nonlinear behavior of stress-strain relation of the material. Since such behavior can be different for different materials, this method gives a large bias error. That is why the major manufacturer of force microscopes and probes, Bruker-Nano started to sell probes with a well-defined geometry for mechanical measurements several years ago. It is good that the authors used the measurements of the geometry in the other way.
3. The use of PDMS (this is a minor issue) Secondly, PDMS is a tricky material. The authors used too high force, which

would correspond to rather high stresses. Since the used macroscopic measurements, they have to compare the stresses used in those measurements with the stresses under the AFM probe. Although elastomers are typically quite linear materials, the difference in the Young's modulus can be very substantial.

4. The use of extremely high indenting forces. Using AFM with such a high force will definitely create an error due to the lateral displacement of the probe. This error is the major reason for the existence of nanoindenters, in which the probe goes vertically down. The authors need to estimate this error if they insist on using a force microscope for such high forces.
5. Taking into account the substrate effect. It has been shown a while ago that the effect of the substrate is substantial. There is nothing new in this observation and the appropriate formulas have to be used [Chiodini, S., et al (2020). Bottom Effect in Atomic Force Microscopy Nanomechanics. *Small*, 16(35), 2000269.].
6. Plastic deformation. "when indenting corneocytes there is plastic deformation." It is a vague statement. Any material can demonstrate plastic deformation starting from a particular stress. In general, it is not clear why the authors need such high stresses, which lead to deformation.
7. Stress-strain curves derived within Herschel-Bulkley model are very interesting. But it is not clear why the other models are worse. An explicit comparison would enhance the manuscript.
8. How long did corneocytes stay out of the human body? They change the internal water content very slowly. It may affect the mechanical properties considerably.

Reviewer #3

(Remarks to the Author)

The work by Ana S. Évora et al presents nanoindentation protocols to measure the mechanical properties of corneocytes, addressing the most important uncertainties of using AFM to describe the mechanical properties of SC cells. This research is very practical and could be a helpful reference for future study. Below I have attached my suggestions and I highly recommend that the authors could revise accordingly:

1. Line 143: 3D projections show two viewing angles (180° apart) of the tip geometry obtained. It seems to be 90 degrees?
2. Line 189: Please further clarify $m(h)$. Is it the power law index as a function of the indentation depth?
3. Line 215-218: The authors have mentioned the relative humidity of their corneocytes sample in the following content. However, it might be better to introduce this condition earlier since the natural moisturizing of corneocytes may also influence authors' assumptions and conclusions? In addition, how much is the maximum indent force applied in determining the recovery of the imprint?
4. In figure 3(a) and (b), the indent force and indent depth are very different when testing different substrates. Did authors have any considerations in setting the maximum indent force and maximum depth?
5. Is the referenced figure in line 247 Figures 4d-h? I think it should be 4c-h.
6. For fig. 4(a), it is still necessary to indicate which line represents approach and which line represents retract. In addition, units are also missing for h and force. For fig.4(a)(b), it might be better if the authors could show the name of h_f , h_e , h_c , h_s , etc., since this figure is pretty far away from the paragraph that mentioned them, which is very inconvenient for readers to quickly understand the scheme.

Version 1:

Reviewer comments:

Reviewer #2

(Remarks to the Author)

The authors only partially addressed the concerns of this reviewer. There are still remaining concerns listed below, and added new ones. Without addressing these concerns, it is not clear if the authors suggest a method that is truly quantitative (in particular, previous concern 4). The main new concern is about quite confusing results summarized in table 2. The authors could estimate the influence of the rigid versus soft substrate. It is not likely to be substantial. This has already been done in the literature. However, the major difference is in the sample preparation. Essentially table 2 says that the suggested method of attachment using xylene doesn't work because it shows a wrong trend of the Young's modulus for different participants. This should be clearly spelled.

Overall, because the use of the sharp probe doesn't allow to calculate the absolute values of the Young's modulus, the method must be really justified by demonstrating its repeatability under different conditions, including different humidities, forces, and the probes of different geometry. This is the cost of using empirical models. So far, it hasn't been done in the submitted manuscript.

On previous concern 1: the authors wrote in the amended text "Spherical colloidal probes are usually used for the mechanical analysis of cells due to their well defined geometry." It is only partially correct. The main reason is to avoid the

nonlinear response of the cell material. Even a sharp probe can have well-defined geometry. Moreover, as was described in the reference suggested previously [P. H. Wu, et al, Nature Methods, 2018, 15, 491] and polymers [M. E. Dokukin et al, Macromolecules, 2012, 45, 4277-4288.], it is sufficient to use a relatively dull but not colloidal probe.

Secondly, the concern about the cost of the dull probes with well-defined spherical geometry, which is suggested by the authors, is superficial. The cost difference is not substantial, and it really disappears when taking into account the longevity of the dull probes compared to the sharp ones. And when speaking about possible medical tests, the cost of the probe will be dissolved in the cost of the equipment and labor anyway.

On previous concerns 1 and 3: The authors say that they can use elastomer PDMS to derive the geometry of the AFM probe. The concern was that the stresses under the AFM probe can be so high that it results in the nonlinear stress-strain response, and as a result, the derived geometry can be incorrect. The authors argued, "The maximum strain of 1.16 in this figure (Fig. S6) corresponds to the maximum displacement of the indenter for the corneocytes so that we could fit the most accurate tip geometry using PDMS." It looks like the authors misunderstood the concern. Certainly, the PDMS response to load a flat punch is linear, even for relatively large strains. The concern was about the nonlinearity of stress and strain under a sharp AFM probe. Both stress and strain must be considered and compared with the flat bench, not only the strain.

On previous concern 2: the authors said, "It is difficult to determine an accurate value of the tip radius of a sharp tip to enable the application of DMT or JKR". Contrary, it is a fairly straightforward procedure using, for example, tip check samples. Moreover, it looks like this is exactly what the authors did to show the probe geometry in figure 1.

On previous concern 4: the authors correctly recognized and calculated the lateral displacement error of the AFM cantilever. In the case of corneocytes, this error was 13 – 25% of the total penetration depth. However, it is unclear how they concluded that this large error results only in 6 – 9% in the calculation of the Young's modulus. In the supplementary materials, in the caption to the supplementary table 3, they mentioned the work of Huang et al. However, that will work doesn't seem applicable to the case of simple measurements done by the authors. Huang et al teach how to calculate the lateral deflection to compensate it using a specially developed AFM mode. Since it was not described in the manuscript, it is unlikely that the authors used that mode. Furthermore, to demonstrate the successful work of the methods, Huang et al used much longer cantilevers compared to the ones used by the authors. Such much longer cantilevers produce lower lateral displacement error. And even then, Huang et al concluded that the error in the modulus can reach 100%. So it is a complete mystery how the authors arrived in only 6 – 9% error.

Without the special AFM mode with lateral compensation described by Huang et al, it does not seem practical to calculate the error of the Young's modulus. To translate the lateral error into the change of the Young's modulus without such a mode, one would need to know the Poisson ratio of the sample material and contact friction between the AFM probe and sample. Neither of them was mentioned by the authors.

On previous concern 8: About humidity control for imaging of corneocytes.

The authors added: "The measurements were done under controlled ambient conditions of 25.5°C and 35% relative humidity, which corresponds to a nominal dry state for these cells, comparable to the conditions on the skin surface in their native state." What is the "nominal dry state"? And if 35% humidity seems to be too low for the "native state"? References to support such an unusual statement are needed.

An additional concern/note: formula S35 is not applicable for the parameters used by the authors because the radius of the probe is smaller than the indentation depth. Therefore, it cannot be considered as indenting by a spherical probe. Strictly speaking, the authors cannot use the formula for the conical probe, S34 because the probe geometry is substantially different from a pure cone. The model for a conespherical indenter should be used.

Reviewer #3

(Remarks to the Author)

The manuscript has been revised according to the suggestions. I recommend publication without further revisions.

Version 2:

Reviewer comments:

Reviewer #2

(Remarks to the Author)

The authors did a very good job improving the manuscript and addressing all of my concerns. Overall, the manuscript is substantially improved now. Obviously, some minor concerns stay, but this is normal for any scientific discussion, and should not prevent this nice work from being published. I am glad to recommend it for publication in its present form.

Response to reviewer comments

Dear reviewers,

We would like to thank you for your comments, suggestions, and detailed review of this manuscript. We have outlined all the changes in the revised manuscript in red font and addressed all the comments below.

Reviewer #1

The paper “Atomic force microscope protocols for characterizing the elastoviscoplastic biomechanical properties of corneocytes” by `Evora et al. addresses very important procedural considerations for cell stiffness measurements. The authors correctly point out that there is considerable variability in data obtained and, although these measurements have been in use for over two decades, there is room for improvement in both data acquisition and analysis approaches. Cell stiffness measurements provide a wealth of information with often significant implications not only in the lab but also in a clinical setting. Such measurements have the potential to diagnose and predict disease outcomes in addition to, of course, furthering our understanding of vital biological processes. Thus, it is important that we get this right and are consistent in our approaches. The authors address tip geometry, substrates and modelling. An initial concern when looking at the data was the use of relatively high indentation forces. In past studies, we have found that using indentation forces in excess of 1 nN could damage living cells and result in indentations that exceeded 1 micron (less than 1 micron ensured good fit with the Hertz model). However, the authors study focuses on corneocytes anuclear, dead cells) that are unlikely to be damaged by higher indentation forces. The authors appropriately address this point in the discussion.

The manuscript offers a thorough investigation into the tip geometry, a very important aspect that must be taken into consideration when selecting the appropriate model to fit cell stiffness data.

The authors conclude that corneocytes exhibit viscoplastic behavior and employ the Herschel-Bulkley material model to fit their data. In summary, this is a thorough investigation into an important approach that needs more consistency. I recommend publication with no revisions.

Response: We appreciate the reviewer’s comments that in this field there is a need to establish a standard protocol for measuring the mechanical properties of corneocytes using standard imaging probes, which can be used to sample both topographical and mechanical properties and to sample a large number of samples in cohort studies.

Reviewer #2

This is a potentially very interesting and important attempt to introduce some standards in the measurements of corneocytes. However, multiple technical issues prevent this reviewer from recommending this work, at least in its present form. Just a few major technical issues:

1. The use of sharp indenter. “wide range of Young’s modulus values reported in the literature..” Since it is a new area, it is plausible to expect that many measurements might be simply incorrect. For example, there are quite a lot of works published demonstrating that the use of excessively sharp probes cannot be utilized to measure the mechanical properties of cells [P. H. Wu, et al, Nature Methods, 2018, 15, 491] and polymers [M. E. Dokukin et al, Macromolecules, 2012, 45, 4277-4288.]. So it is not clear why the author didn’t use quite generally accepted these days spherical indenter as with well-defined geometry.

Response: We agree that sharp indenters can create problems when measuring mechanical properties. The protocol was developed to determine the potential of corneocytes as a biomarker of, for example, early-stage pressure ulcers. The aim is to study much larger cohorts with a greater number of samples to be analysed, which would involve the use of a large number of probes. Indenters with well-defined geometries are considerably more expensive and budget constraints would limit the number of samples that could be evaluated. Moreover, an advantage of sharp indenters is that they allow the cells to be imaged followed by force spectroscopy measurements on zoomed regions of the cells. We should have made this clear in the paper and have re-written the Abstract. In addition, the following text to this effect has been added in the Introduction:

Added to the Introduction:

Spherical colloidal probes are usually used for the mechanical analysis of cells due to their well defined geometry.¹⁵ However, one of the main advantages of AFM is the coupling of topographical and biomechanical analysis that can be achieved with standard imaging probes, also known as sharp AFM tips. This is particularly important for assessing how the surface properties of corneocytes correlate with their mechanical behaviour. In fact, the presence of circular nano-objects at the surface of skin cells has been associated with certain skin conditions such as Atopic Dermatitis¹¹, but there has not been any attempts to correlate these maturation properties with the stiffness of skin cells. Furthermore, indenters with well-defined geometries are considerably more expensive than standard imaging probes, which may limit studies on large cohorts for clinical applications.

With reference to the Dokukin et al paper ¹, they concluded that when using excessively sharp probes, the skin-effect originates mainly due to the nonlinearity of the stress–strain relationship and if the probe–surface adhesion is not considered. It is difficult to determine an accurate value of the tip radius of a sharp tip to enable the application of DMT or JKR, which may make some contribution to the skin-effect observed in this paper. However, we attenuated the adhesion for the PDMS measurements by using aqueous SDS and the contribution for the skin cells was relatively small. As the reviewer stated in point 3, elastomers are typically quite linear materials. In response to the reviewers comments we have made the following changes. Fig. S6 is a plot of the micromanipulator force as a function of displacement, which is linear showing that the Young’s modulus of the PDMS is constant for the strain range considered. The maximum strain of 1.16 in this figure (Fig. S6) corresponds to the maximum displacement of the indenter for the corneocytes so that we could fit the most accurate tip geometry using PDMS. This is detailed in point 3.

Added to Supplementary Information S5:

Flat punch theory (Eq. S29) was applied to an indentation depth that is equivalent to the maximum representative strain calculated for a depth of 70 nm, which is the average indentation depth applied to the corneocytes. To calculate the representative elastic strain for the indentation of the PDMS by the AFM tip, an expression was derived for a power law indenter based on the force, F , as a function of indentation depth, h ⁵:

$$F = E^* \frac{2n}{n+1} \left(\frac{1}{\kappa c} \right)^{1/n} h^{(n+1)/n} \quad \text{S22}$$

where n , c , and κ are the tip geometry parameters, as described in S1. Then, the stress can be described as a function of h and a as:

$$\sigma = \frac{F}{\pi a^2} = E^* \frac{2n}{n+1} \left(\frac{1}{\kappa c} \right)^{1/n} \frac{h^{(n+1)/n}}{\pi a^2} \quad \text{S23}$$

and h , the indentation depth, may be defined as a function of the contact radius, a :

$$h(a) = \kappa(n)ca^n \quad \text{S24}$$

Using Eqs. S23 and S24:

$$\sigma = \frac{E}{(1-\nu^2)} \frac{2n}{\pi(n+1)} \kappa ca^{n-1} = \varepsilon E \quad \text{S25}$$

Finally, the strain for the AFM probe can expressed as:

$$\varepsilon = \frac{1}{(1-\nu^2)} \frac{2n}{n+1} \frac{\kappa c a^{n-1}}{\pi} \quad \text{S26}$$

which reduces to the following expression:

$$\varepsilon = \frac{2n}{\pi(1-\nu^2)(n+1)} \frac{h}{a} \quad \text{S27}$$

At $h \approx 70$ nm, the strain was calculated to be ~ 1.16 . This is generally termed a representative strain since the strain field under an indenter is not uniform and thus cannot be compared directly with, for example, that corresponding to simple uniaxial extension.

Therefore, flat punch theory was applied to a depth up to this strain ($h \approx 15$ μm in Figure S6) for the loading curve obtained using the micromanipulator and calculated using Eq. S28 (Eq. S27 with $n \gg 1$ and $\nu = 0.5$). Figure S6 represents a typical micromanipulator loading curve as well as the strain.

$$\varepsilon = 0.85 \frac{h}{a} \quad \text{S28}$$

Fig. S1. Force-depth and strain-depth curves for PDMS. Force and strain-displacement curve up to 20 μm indentation depth obtained using a flat punch and a micromanipulation technique.

The Young's modulus of PDMS was then calculated on the basis of a rigid flat punch indenting an isotropic linear elastic half-space (flat-punch theory)⁷:

$$F = 2RE^*\delta \quad \text{S29}$$

where F is applied force, R is the radius of the flat punch (12 μm), and δ is the displacement and E^* of the PDMS is given by the following expression:

$$E^* = \frac{E}{1-\nu^2} \quad \text{S30}$$

where E and ν are the Young's and Poisson's ratio of the PDMS, which was taken as 0.5 since it is considered incompressible. Fig. S6 shows that the PDMS is linear elastic in the range of strains relevant to the AFM measurements of this elastomer.

2. The use of a reference material to restore the probe geometry. "a silicone elastomer reference material for determining the effective indenter geometry". The method which was recognized about 10 years ago as putting "dust onto the carpet". When using a sharp indenter, the recovered geometry of the indenter will depend on the nonlinear behaviour of stress-strain relation of the material. Since such behaviour can be different for different materials, this method gives a large bias error. That is why the major manufacturer of force microscopes and probes, Bruker-Nano started to sell probes with a well-defined geometry for mechanical measurements several years ago. It is good that the authors used the measurements of the geometry in the other way.

Response: The reasons for using a sharp indenter were explained in point 1 together with the evidence for the linear elastic behaviour of the PDMS for the range of strains corresponding to that of the AFM measurements of the PDMS. Such probes do not have well-defined axisymmetric geometries so that the use of a reference material has the advantage of determining the effective tip geometry.

3. The use of PDMS (this is a minor issue) Secondly, PDMS is a tricky material. The authors used too high force, which would correspond to rather high stresses. Since the used macroscopic measurements, they have to compare the stresses used in those measurements with the stresses under the AFM probe. Although elastomers are typically quite linear materials, the difference in the Young's modulus can be very substantial.

Response: Since the Young's modulus of the PDMS is orders of magnitude less than that of the corneocytes, and thus that of the stresses, we have ensured that the maximum strain applied in the micromanipulator measurements was comparable to those applied in the AFM measurements involving both the PDMS as explained in point 1. Also, as explained previously, in the supplementary Information, a representative force curve (Figure S6) shows a linear relationship between the force and displacement for the flat punch used in the micromanipulator measurements, which is consistent with linear elastic behaviour.

4. The use of extremely high indenting forces. Using AFM with such a high force will definitely create an error due to the lateral displacement of the probe. This error is the major reason for the existence of nanoindenters, in which the probe goes vertically down. The authors need to estimate this error if they insist on using a force microscope for such high forces.

Response: We appreciate this comment since the reviewer is correct that the high forces create lateral displacement of the AFM probe. Huang et al ². noted that high forces can generate lateral displacement of the AFM probe. The lateral deflection of the probe (Δx) at F_{max} can be calculated considering that ⁸:

$$\Delta x = \Delta x_1 + \Delta x_2 \quad 1$$

where:

$$\Delta x_1 = \text{cantilever deflection}_{F_{max}} \times \tan \theta \quad 2$$

Where the cantilever tilt angle, $\theta = 10^\circ$ (for Nanowizard 4, Bruker, JPK BioAFM, Berlin, Germany) and

$$\Delta x_2 = \text{cantilever deflection}_{F_{max}} \frac{3 \times \text{AFM tip height}}{2 \times \text{cantilever length}} \quad 3$$

where AFM tip height and cantilever length are given by the supplier as 15 and 117 μm , respectively. Using Eqs. (2) and (3), it was possible to calculate the potential lateral deflection for PDMS, which was less than 1% of the total penetration depth. Therefore, the effect of any lateral displacement on the AFM data for the PDMS was not significant.

For the case of corneocytes, Δx was observed to be in the range 13–25% of the total penetration depth. This resulted in an error range for the calculated Young's modulus of 6–9 %. To account for this error, a table (S3) was added to the supplementary information of this draft including the error associated to the Young's modulus of corneocytes calculated for each participant. In practice, we would expect this effect to be at least partially mitigated by compensating errors in the use of a rigid substrate to calibrate the deflection sensitivity of the cantilever, and a reference sample of known elastic modulus to calibrate the probe shape.

There are methods that have been developed for minimising lateral displacement errors for AFM nanoindentation. They include lateral motion compensation, the use of non-deflecting cantilevers and reduced cantilever tilt angles^{3,4,5}. Unfortunately, it was not possible to implement them on our AFM.

ADDED to section - Nanoindentation and stress relaxation of corneocytes:

Furthermore, the forces used in this study for the measurements performed on the cells attached to a glass slide (~ 2 µN), may create an error due to potential lateral deflection of the AFM probe. The maximum mean error in the Young's moduli was estimated to be in the range 6–9 % (Table S3).

ADDED to Supplementary Information:

S6 Lateral deflection of AFM probe

Huang et al⁸ suggested that high forces can generate lateral displacement of the AFM probe. They proposed that the lateral deflection of the probe (Δx) at F_{max} can be calculated as follows⁸:

$$\Delta x = \Delta x_1 + \Delta x_2 \quad \text{S31}$$

where:

$$\Delta x_1 = \text{cantilever deflection}_{F_{max}} \cdot \tan \theta \quad \text{S32}$$

where the cantilever tilt angle, $\theta = 10^\circ$ (for Nanowizard 4, Bruker, JPK BioAFM, Berlin, Germany) and

$$\Delta x_2 = \text{cantilever deflection}_{F_{max}} \cdot \frac{3 \times \text{AFM tip height}}{2 \times \text{cantilever length}} \quad \text{S33}$$

where the AFM tip height and cantilever length are given by the supplier as 15 and 117 µm, respectively. Using Eqs. (S32) and (S33), it was possible to calculate the potential lateral deflection for PDMS, which was less than 1% of the total penetration depth. Therefore, the effect of any lateral displacement on the AFM data for the elastomer was not significant.

For the case of corneocytes, Δx was observed to be in the range 13–25% of the total penetration depth. This resulted in an error range for the calculated Young's modulus of 6–9 % (Table S3). This error was calculated after compensating the values of indentation depth with a corrected vertical deflection value, i.e., based on the cantilever stiffness and on Δx .

However, in practice this error is likely to be partially compensated for by the use of a rigid substrate to calibrate the deflection sensitivity of the cantilever, and a reference sample of known elastic modulus to calibrate the probe shape. Moreover, the above calculation assumes that there is not any resistance to the lateral motion but in practice it would require a lateral deformation of the corneocytes.

The phenomenon of lateral displacement errors is a well-known issue with AFM nanoindentation and could be addressed using strategies already established for minimising its effects. These include lateral motion compensation, the use of non-deflecting cantilevers and reduced cantilever tilt angles ^{9, 10, 11}.

Table S3. Error calculated for the values of corneocyte Young's modulus considering the potential lateral deflection of the AFM probe. Percentage of error is shown as mean \pm 1 SD (for 5 cells per participant and 64 force curves per cells). Error was calculated based on the work by Huang et al. on lateral forces during AFM indentation ⁸.

Participant	Young's modulus error (%)
1	6.5 \pm 1.6
2	9.6 \pm 4.8
3	9.4 \pm 4.2

5. Taking into account the substrate effect. It has been shown a while ago that the effect of the substrate is substantial. There is nothing new in this observation and the appropriate formulas have to be used [Chiodini, S., et al (2020). Bottom Effect in Atomic Force Microscopy Nanomechanics. Small, 16(35), 2000269.].

Response: We agree that the substrate effect in nanoindentation is well known. However, the authors wanted to emphasize that previous experiments on corneocytes, that are generally collected via tape stripping, did not account for the effect of soft substrates. Although this has been shown for soft cells on hydrogels (Rheinlaender et al. 2020) ⁶, this has not been considered for corneocytes using tape as the measurement substrate. We recognize that this is even more important in the case of corneocytes due to their high stiffness compared to the tape adhesive.

Moreover, we wanted to point out that the effect of hard substrates (in this case, a glass slide) was also considered by reducing the indentation depths to $< 10\%$ of the cell thickness. On the basis of Garcia et al. ⁷, we show that, using the generalized expressions for both a paraboloid and a cone, we obtain:

- approximating our indenter as a cone with a half-angle $\theta \sim 22^\circ$ (measured from an SEM image), the second and higher order terms in Eq (14a) of their paper are $\ll 1$, and so can be ignored.
- approximating our indenter as a paraboloid with a radius $R \sim 50$ nm, the second and higher order terms in Eq (16a) of their paper are also $\ll 1$, and so can be ignored.

Furthermore, Dimitriadis et al. ⁸ states that to avoid the effects of the substrate:

$$\delta \text{ (indentation depth)} < 0.1 h \text{ (sample thickness)} \quad (4)$$

The average corneocyte cell height is 0.8 ± 0.1 μm and we performed experiments with maximum indentation depths, δ , of 70 ± 10 nm, thus:

$$\delta < 0.089 h$$

ADDED to Supplementary Information:

S7 Effect of a stiff substrate

The effect of hard substrates (in this case, a glass slide) was considered by reducing the indentation depths to $< 10\%$ of the cell thickness. Based on Garcia et al. ⁹, and taking the generalized expressions for conical and paraboloid geometries:

- If approximating the AFM indenter to a cone with a half-angle $\theta \sim 22^\circ$ (measured from an SEM image), and considering ⁹:

$$F_{\text{cone}} = F_0 \left[\frac{1}{\text{sample thickness}^0} + \frac{0.721h \tan \theta}{\text{sample thickness}} + \frac{0.65h^2 \tan^2 \theta}{\text{sample thickness}^2} + \frac{0.491h^3 \tan^3 \theta}{\text{sample thickness}^3} + \frac{0.2251h^4 \tan^4 \theta}{\text{sample thickness}^4} \right] \quad \text{S34}$$

where h is the maximum indentation depth taken as an average of 70 ± 20 nm. For an average sample thickness of 0.8 μm , the sum of the second and higher order terms in Eq. S34 is 0.092 , i.e., $\ll 1$.

- If approximating the AFM indenter to a paraboloid:

$$F_{\text{paraboloid}} = F_0 \left[\frac{1}{\text{sample thickness}^0} + \frac{1.133\sqrt{hR}}{\text{sample thickness}} + \frac{1.497hR}{\text{sample thickness}^2} + \frac{1.469hR\sqrt{hR}}{\text{sample thickness}^3} + \frac{0.755(h^2R^2)}{\text{sample thickness}^4} \right] \quad \text{S35}$$

where h is the maximum indentation depth taken as an average of 70 ± 20 nm and R is the approximate tip radius taken as 53.2 ± 6.9 nm (see Fig. 1). Then, the sum of the second and higher order terms in Eq. S35 is 0.023, i.e., $\ll 1$.

Moreover, Dimitriadis et al. ¹⁰, states that to avoid the effects of the substrate:

$$h < 0.1 \times \text{sample thickness} \quad (\text{S36})$$

The average corneocyte cell height is 0.8 ± 0.1 μm and we performed experiments with maximum indentation depths, h , of 70 ± 10 nm, thus:

$$h < 0.089 \text{ sample thickness} \quad (\text{S37})$$

6. Plastic deformation. "when indenting corneocytes there is plastic deformation." It is a vague statement. Any material can demonstrate plastic deformation starting from a particular stress. In general, it is not clear why the authors need such high stresses, which lead to deformation.

Response: The Young's moduli of the corneocytes found in the current work (0.4–1 GPa) are consistent with the values found for stratum corneum (Park and Baddiel 1972) ⁹. Furthermore, they are similar to those of glassy polymers, which have been studied using AFM nanoindentation with much higher forces and sharp tips ^{10,11,12}.

The use of high forces is then justified by the need for relevant indentation depths, so that the bulk mechanical properties of corneocytes could be calculated. In fact, corneocytes can be considered as a composite material: a keratin matrix surrounded by a cornified envelope ($\sim 5\text{--}20$ nm ^{13, 14}). As such, an indentation depth of around 70 nm would allow the study of the bulk mechanical behaviour of these cells, which was only achieved at the forces used. Under these conditions, the strain imposed by the indenter geometry exceeds the corneocyte's yield strain enabling viscoplastic behaviour to be characterised.

7. Stress-strain curves derived within Herschel-Bulkley model are very interesting. But it is not clear why the other models are worse. An explicit comparison would enhance the manuscript.

Response: The Herschel-Bulkley model is simply an empirical relationship that describes the post-yield experimental data i.e., it is no different in kind, for example, from that which

describes the relationship between the stress and the strain rate for a Newtonian liquid. That is, it is not based a fundamental theory and thus it is not obvious what other models could be used as a comparison. There are models for elastoplasticity but, to the authors knowledge, not for elastoviscoplasticity.

8. How long did corneocytes stay out of the human body? They change the internal water content very slowly. It may affect the mechanical properties considerably.

Response: Corneocyte samples were frozen after collection. To perform measurements, the samples were allowed to equilibrate to the ambient temperature in a temperature and humidity-controlled laboratory, and xylene extraction was performed overnight. Thus, the cells were effectively out of the human body and at ambient conditions for about 12 h before measurements. We agree that the water content can affect the mechanical properties considerably, but all samples were treated in the same way and analysed following the same timeline. The effects of water content on corneocyte mechanics were not the aim of the current work and will be analysed in a future study.

References

1. Dokukin ME, Sokolov I. On the Measurements of Rigidity Modulus of Soft Materials in Nanoindentation Experiments at Small Depth. *Macromolecules* **45**, 4277-4288 (2012).
2. Huang L, Meyer C, Prater C. Eliminating Lateral Forces During AFM Indentation. *Journal of Physics: Conference Series* **61**, 805 (2007).
3. Hoh J. H., Engel A., Friction effects on force measurements with an atomic force microscope, *Langmuir* **9(11)** 3310-3312 (1993).
4. Vanlandingham M. R., McKnight S. H., Palmese, G. R. Eduljee R. F., Gillespie J. W., McCulough J. R. R. L. Relating elastic modulus to indentation response using atomic force microscopy. *J Mater Sci Lett* **16**, 117–119 (1997).
5. Clifford C. A., Seah M. P., Quantification issues in the identification of nanoscale regions of homopolymers using modulus measurement via AFM nanoindentation, *Appl. Surf. Sci.* **252(5)** 1915-1933 (2005).
6. Rheinlaender J, *et al.* Cortical cell stiffness is independent of substrate mechanics. *Nat Mater* **19**, 1019-1025 (2020).
7. Garcia PD, Garcia R. Determination of the Elastic Moduli of a Single Cell Cultured on a Rigid Support by Force Microscopy. *Biophys J* **114**, 2923-2932 (2018).
8. Dimitriadis EK, Horkay F, Maresca J, Kachar B, Chadwick RS. Determination of Elastic Moduli of Thin Layers of Soft Material Using the Atomic Force Microscope. *Biophys J* **82**, 2798-2810 (2002).
9. Park A, Baddiel C. Rheology of stratum corneum-I: A molecular interpretation of the stress-strain curve. *J Soc Cosmet Chem* **23**, 3-12 (1972).
10. Cappella B, Kaliappan SK, Sturm H. Using AFM Force–Distance Curves To Study the Glass-to-Rubber Transition of Amorphous Polymers and Their Elastic–Plastic Properties as a Function of Temperature. *Macromolecules* **38**, 1874-1881 (2005).
11. Tranchida D, Sperotto E, Chateauminois A, Schönherr H. Entropic Effects on the Mechanical Behavior of Dry Polymer Brushes During Nanoindentation by Atomic Force Microscopy. *Macromolecules* **44**, 368-374 (2011).
12. Chaurasia SS, *et al.* Effect of Fibrin Glue on the Biomechanical Properties of Human Descemet's Membrane. *PLoS One* **7**, e37456 (2012).
13. Eckert RL, Sturniolo MT, Broome A-M, Ruse M, Rorke EA. Transglutaminase Function in Epidermis. *J Invest Dermatol* **124**, 481-492 (2005).
14. Milani P, Chlasta J, Abdayem R, Kezic S, Haftek M. Changes in nano-mechanical properties of human epidermal cornified cells depending on their proximity to the skin surface. *J Mol Recognit* **31**, e2722 (2018).

Reviewer #3:

The work by Ana S. Évora et al presents nanoindentation protocols to measure the mechanical properties of corneocytes, addressing the most important uncertainties of using AFM to describe the mechanical properties of SC cells. This research is very practical and could be a helpful reference for future study. Below I have attached my suggestions and I highly recommend that the authors could revise accordingly:

1. Line 143: 3D projections show two viewing angles (180° apart) of the tip geometry obtained. It seems to be 90 degrees?

Response: We can confirm that the projections have been rotated 180°.

2. Line 189: Please further clarify $m(h)$. Is it the power law index as a function of the indentation depth?

Response: Yes, $m(h)$ is the power law as a function of the indentation depth. This has been included in the main text.

3. Line 215-218: The authors have mentioned the relative humidity of their corneocytes sample in the following content. However, it might be better to introduce this condition earlier since the natural moisturizing of corneocytes may also influence authors' assumptions and conclusions? In addition, how much is the maximum indent force applied in determining the recovery of the imprint?

Response: This is a good point; we have altered the text so that we mention the relative humidity of the experiments.

Added text to “Nanoindentation and stress relaxation of corneocytes” section:

The measurements were done under controlled ambient conditions of 25.5°C and 35% relative humidity, which corresponds to a nominal dry state for these cells, comparable to the conditions on the skin surface in their native state.

Moreover, the maximum indent force applied to determine recovery imprint is 2 μN . This information was added to Figure 3 legend: “of cells in a glass slide with a force of 2 μN ”.

4. In figure 3(a) and (b), the indent force and indent depth are very different when testing different substrates. Did authors have any considerations in setting the maximum indent force and maximum depth?

Response: The force applied in force spectroscopy was set accordingly to displacements no greater than 10% of the cell thickness. Following the work of Dimitriadis et al. ⁵, that states that, to avoid the effect of the substrate,

$$\delta \text{ (indentation depth)} < 0.1 h \text{ (sample thickness)}$$

And considering that the average corneocyte cell height is $0.8 \pm 0.1 \mu\text{m}$, we performed experiments with maximum indentation depths, δ , of $60 \pm 20 \text{ nm}$, and so

$$\delta < 0.089$$

5. Is the referenced figure in line 247 Figures 4d-h? I think it should be 4c-h.

Response: This is correct, it should be 4c-h and it is now corrected in the main text.

6. For fig. 4(a), it is still necessary to indicate which line represents approach and which line represents retract. In addition, units are also missing for h and force. For fig.4(a)(b), it might be better if the authors could show the name of hf, he, hc, hs, etc., since this figure is pretty far away from the paragraph that mentioned them, which is very inconvenient for readers to quickly understand the scheme.

Response: Thank you for noting this. We have now corrected figure 4(a) and named all the heights presented in figures 4(a) and 4(b).

Atomic force microscope protocols for characterising the elastoviscoplastic biomechanical properties of corneocytes (COMMSBIO-23-1499A)

Dear Reviewers,

We would like to thank you for your constructive comments, suggestions, and detailed reviews of our revised manuscript, as detailed in the email from the editors on 25th July 2023. All 8 comments on this version from Reviewer #2 (*in italics*) requiring a response are addressed below. The resulting changes in the revised main text and Supplementary Information (SI) are marked in **blue** font. In addition, to aid symbol identification, we have also included a nomenclature table as an appendix in the SI. This has also enabled the equations in SI Sections S8 and S9 to be expressed in a more succinct form. Two improvements (in **orange**) have also been made to the main text version submitted on the 21st June 2023, based on the most recent literature:

Discussion line 392: The current results are also consistent with those recently published by Boonpuek et al.⁴⁵, who presented values of Young's modulus in the range 0.6–1.0 GPa for single corneocytes attached to an aluminium plate. The results are based on the JKR model for the adhesive surface energy of elastic solid-solid contact between two spheres.

Discussion line 455: The current methodology has been used to study the effects of water activity on the mechanical properties of corneocytes and to describe the mechanical behaviour of corneocytes⁵⁰ collected from several anatomical sites⁵¹.

Reviewer #2

The authors only partially addressed the concerns of this reviewer. There are still remaining concerns listed below and added new ones. Without addressing these concerns, it is not clear if the authors suggest a method that is truly quantitative (in particular, previous concern 4).

- 1. The main new concern is about quite confusing results summarized in table 2. The authors could estimate the influence of the rigid versus soft substrate. It is not likely to be substantial. This has already been done in the literature. However, the major difference is in the sample preparation. Essentially table 2 says that the suggested method of attachment using xylene doesn't work because it shows a wrong trend of the Young's modulus for different participants. This should be clearly spelled.*

Response: We understand the reviewer's comments regarding Table 2, which requires addressing in the manuscript. The effect of the substrate for AFM measurements is an important point, especially considering the difference in stiffness between the substrate (tape) and corneocytes.

As mentioned in the Introduction and in the Materials and Methods sections, the adhesive of regular tapes is usually composed of polyacrylates. They are relatively soft materials with much smaller values of Young's moduli (e.g. 0.45 MPa¹) than those typically reported for relatively dry stratum corneum (1–2 GPa, Park and Baddiel reference 32 in the main text). In Table 2, there are, indeed, conflicting trends between the ratios of the measurements on tape and those on a glass slide, particularly for Participant 2. However, the authors disagree that this is an indication that the xylene extraction method does not work. We believe that the results between tape and glass should not be directly compared. The forces used to "indent" the cells attached to the tape are about a factor of 8 times less than those used to indent the skin cells attached to the glass slide, which is consistent with the cells being pushed down into the soft tape substrate. This is supported by the work of Rheinlaender et al.² for other cell types on soft substrates. Moreover, in the SI (Section S4), we present data from FTIR spectroscopy, showing no apparent chemical/structural differences between cells before and after xylene extraction, which indicates that major chemical changes have not occurred to the corneocytes upon xylene extraction. Consequently, we suggest that the force curves obtained for cells attached to the tape are mostly dependent on the deformability of the substrate, rather than on cell indentation. Furthermore, the tape adhesive is characterized by a considerable roughness and a variation of the thickness that has a length scale much greater than that of the cells. This influences the contact between a cell and the tape and the extent of indentation of a cell into the adhesive. An additional figure showing this has been included in the SI (see Figure S8, below) to help reinforce the point.

We consider that the effect of the soft substrate on the force curves cannot be ignored, and any conclusions should not be based on them. Although the effect of a soft substrate on cell stiffness has been investigated by Rheinlaender et al.² for other cell types on soft substrates, we could not use their model to correct our data, considering the complex geometry of our probe, the elastoviscoplastic behaviour of corneocytes and the wide variation in the thickness of the adhesive. This is mentioned in the Introduction (line 86), but it is now emphasised in the text, as follows:

Added to the Results section “Nanoindentation and stress relaxation of corneocytes”:

Line 267: Considering the significant effect of the soft substrate (tape) on the nanoindentation results, the values of Young’s modulus obtained for the cells attached to the tape should not be compared with those of cells attached to the glass slide. In fact, as previously studied for fibroblasts on soft gels, the effect of substrate deformability on cell stiffness measurements can be considerable¹⁵. The adhesives that constitute regular tapes, such as the one used in the current study, belong to the family of polyacrylates, which are characterized by relatively small values of Young’s modulus ($\sim 2 \times 10^4 - 5 \times 10^5$ Pa)³¹, when compared to comparatively stiff SC in a relatively dry state³². Therefore, the effect of the deformability of the adhesive cannot be excluded for indentations of cells attached to a tape. Due to the complex AFM tip geometry, the wide variation of the thickness of the adhesive, and the mechanical behaviour of the corneocytes described in the current study, it was not possible to correct this effect using, for example, the CoCS model¹⁵. That the trends in the Young’s moduli corresponding to the two substrates for the three participants are not systematic, is a result of a significant variation in the thickness and surface topography of the adhesive (Supplementary Figure S8).

Added to Supplementary Information:

S7 Topography of the soft substrate - tape

Fig. S1. Optical microscope images of Sellotape showing the two main parts of the tape: a backing film known as the carrier and a pressure-sensitive adhesive usually composed of polyacrylate derivatives. The adhesive layer is characterized by a certain roughness and variable thickness that will influence the interaction force with a cell.

2. Overall, because the use of the sharp probe doesn't allow to calculate the absolute values of the Young's modulus, the method must be really justified by demonstrating its repeatability under different conditions, including different humidities, forces, and the probes of different geometry. This is the cost of using empirical models. So far, it hasn't been done in the submitted manuscript.

Response: We agree that we did not provide evidence of the repeatability of this methodology. We have now included a section in the SI (Section S6) that addresses this issue by describing some results from nanoindentation on polymethyl methacrylate (PMMA). A new paragraph noting this is also included in the main text. In this work, two different cantilevers were used, and the resulting Young's moduli data were compared for a maximum force of 3.40 μN . Moreover, for one of the cantilevers, the results are compared for different force setpoints (2.5, 3.40, 4.15 and 6.30 μN). The geometry of both cantilevers was calibrated using the described method, i.e., using PDMS as a reference elastomer. The agreement between the results is strong evidence of the applicability of this method and that the variability based on 3 x 64 indents for each set point is relatively small. Moreover, the average value of the Young's moduli obtained (2.5–2.9 GPa) is close to the reported range for PMMA of 2.8–3.3 GPa³ (See Table S3 below).

The effects of water activity on the mechanical properties of corneocytes has also been studied and published elsewhere using the methodology described in this paper⁴. Volar forearm and medial heel corneocytes collected from three participants presented Young's moduli of ~ 620 and ~ 564 MPa in the relatively dry state (35% RH), respectively, and were reduced to ~ 1.5 MPa in the wet state, i.e., incubated in water. These values, which are given in Table 1 below, are consistent with those reported by Park and Baddiel (reference 32 in main text) using an extensometer for SC isolated from pig ears. The Young's modulus was found to be ~ 2 GPa, at 30% RH and 25 °C, while at 100% RH, it decreased to ~ 3 MPa.

Table 1. Mechanical properties of forearm and medial cells with hydration reproduced as in published paper⁴. There were 3 participants and 6 cells per participant, mean \pm SD.

Hydration level		E (MPa)	
		Forearm	Medial Heel
Air (RH)	0.35	620 \pm 320	564 \pm 59
Glycerol 85% (A_w)	0.42	6.9 \pm 2.0	6.2 \pm 0.9
Glycerol 70% (A_w)	0.62	2.1 \pm 1.1	3.3 \pm 1.5
Glycerol 50% (A_w)	0.80	1.8 \pm 0.2	1.8 \pm 0.9
H ₂ O (A_w)	1.00	1.4 \pm 0.1	1.5 \pm 0.8

Added to main text (line 218):

To validate this calibration method, a sample of polymethyl methacrylate (PMMA) sheet (Merck KGaA, Darmstadt, Germany) was indented with two different AFM probes and with different force setpoints. The values of elastic modulus obtained from the indentation data were satisfactorily close to those in the literature (see Supplementary Section 6).

Added to Supplementary Information:

S6 AFM nanoindentation of polymethyl methacrylate (PMMA)

A specimen of smooth cast PMMA sheet (Merck KGaA, Darmstadt, Germany) was used to demonstrate the utility and validity of the current tip calibration method. Nanoindentation experiments were performed using two different probes (cantilever A and B) and a range of maximum force setpoints for cantilever A (2.50, 3.40, 4.15 and 6.30 μN). The geometry of both probes was initially calibrated using the described methodology, i.e., by performing nanoindentation on the reference PDMS elastomer from which a tip contact radius function was obtained (Fig. S7). The measurements followed the described protocol, i.e., a total of 3 regions per cantilever and force setpoint with 64 loading and unloading force curves collected in a $5 \times 5 \mu\text{m}$ region at a velocity of $0.5 \mu\text{m/s}$. This included a dwell time of 4 s after loading to allow any viscous components to relax at the maximum force. The Oliver-Pharr method was employed to calculate the Young's moduli (assuming that $\nu = 0.35$) from the unloading curves and the results are shown in Table S3. There is a relatively small variability between the data obtained from the 3×64 indents for each set point and close agreement between the five results. Moreover, the average values of the Young's moduli obtained (2.5–2.9 GPa) are satisfactorily close to the reported range for cast PMMA of 2.8–3.3 GPa⁸.

Fig. S2. The tip contact radius functions for two probes (Cantilever A and B) used for the nanoindentation of PMMA (average of three regions and 64 loading curves), which were obtained from a fit of $a(h_c)$ values from $h_c = 5$ to 150 nm.

Table S2. Young's moduli of PMMA obtained using two AFM probes and different maximum force setpoints.

Cantilever	Force (μN)	Indentation depth (nm)	Young's modulus (GPa)
A	2.50	18 ± 3	2.7 ± 0.4
	3.40	24 ± 4	2.6 ± 0.4
	4.15	32 ± 3	2.9 ± 0.4
	6.30	38 ± 3	2.5 ± 0.4
B	3.40	29 ± 10	2.7 ± 0.5

3. *On previous concern 1: the authors wrote in the amended text “Spherical colloidal probes are usually used for the mechanical analysis of cells due to their well-defined geometry.” It is only partially correct. The main reason is to avoid the nonlinear response of the cell material. Even a sharp probe can have well-defined geometry. Moreover, as was described in the reference suggested previously [P. H. Wu, et al, Nature Methods, 2018, 15, 491] and polymers [M. E. Dokukin et al, Macromolecules, 2012, 45, 4277-4288.], it is sufficient to use a relatively dull but not colloidal probe.*

Secondly, the concern about the cost of the dull probes with well-defined spherical geometry, which is suggested by the authors, is superficial. The cost difference is not substantial, and it really disappears when taking into account the longevity of the dull probes compared to the sharp ones. And when speaking about possible medical tests, the cost of the probe will be dissolved in the cost of the equipment and labor anyway.

Response: We agree that the quoted sentence is misleading. It is the case that colloidal probes, apart from having the advantage of a well-defined spherical geometry, which allows the assumption of a Hertzian or DMT/JKR contact, are also used to avoid damaging the cell material. In fact, this is directly related to the high stresses generated by sharp probes, as mentioned by the reviewer, and investigated in the mentioned literature. We have now edited the main text to take this into account.

The use of sharp probes has the great advantage of cell imaging, which allowed us to map the surface topography of the corneocytes, at greater spatial resolution compared to dull probes, followed by their mechanical analysis. The analysis of the topography of the cells provides important information about their structure^{5, 6, 7} particularly coupled with the mechanical studies. Moreover, the use of sharp probes in the current study allowed us to study the non-linear elastic properties of corneocytes by applying stresses that satisfied the yield criterion.

Concerning the second point, we have removed the mention of the relative cost of different probes in the abstract.

Added to the introduction (line 95):

Spherical colloidal probes are usually used for the mechanical analysis of cells due to their well-defined geometry and the need to avoid the non-linear response of the cell material that can result in cellular damage¹⁵. However, by using sharp probes for nanoindentation it is possible to apply sufficient stresses that cause plastic deformation, and thus allow the elastoplastic properties to be characterised for cells such as corneocytes, which are more

resilient to such damage. Furthermore, one of the main advantages of AFM is the coupling of topographical and biomechanical analyses that can be achieved with standard imaging probes, also known as sharp AFM tips. This is particularly important for assessing the correlation of the surface properties of corneocytes with their mechanical behaviour. In fact, the presence of circular nano-objects at the surface of skin cells has been associated with certain skin conditions such as atopic dermatitis¹¹, but there have not been any attempts to correlate these maturation properties with the stiffness of the skin cells.

Modified the abstract (line 35):

It involved the use of standard sharp imaging AFM probes, which have the advantages of being able to capture the initial cell topography and select specific indentation locations for measuring the mechanical properties without requiring probe changes.

4. *On previous concerns 1 and 3: The authors say that they can use elastomer PDMS to derive the geometry of the AFM probe. The concern was that the stresses under the AFM probe can be so high that it results in the nonlinear stress-strain response, and as a result, the derived geometry can be incorrect. The authors argued, "The maximum strain of 1.16 in this figure (Fig. S6) corresponds to the maximum displacement of the indenter for the corneocytes so that we could fit the most accurate tip geometry using PDMS." It looks like the authors misunderstood the concern. Certainly, the PDMS response to load a flat punch is linear, even for relatively large strains. The concern was about the nonlinearity of stress and strain under a sharp AFM probe. Both stress and strain must be considered and compared with the flat bench, not only the strain.*

Response: The authors agree that the problem of high stresses and strains associated with sharp probes may prove to be a problem in terms of a non-linear response and that it may generate a skin-effect, as well as potential hysteresis artefacts, as that present for the polymers investigated by Dokukin and Sokolov⁸. To ensure that the measurements involving the PDMS were in the linear elastic regime, we calculated the maximum stress and strain associated with a sharp probe for a maximum indentation depth of 70 nm, which is approximately 3.3 MPa and 1.16, respectively. In fact, these values are within the linear elastic region for this specimen of PDMS, as determined by the micromanipulator results. We have updated the relevant section in the SI and added a stress-strain plot to Fig. S6 make this clearer (see Fig. S6b below).

For the same 70 nm indentation depth, the strain for corneocytes using AFM was ~ 0.8 , which corresponded to stresses of ~ 150 MPa, which more than sufficient to induce plastic deformation.

Added to the SI (section S5):

For a 20 μm indentation depth with $a = 12$ μm , Eq. S30 gives $\varepsilon = 1.4$. Since the force-depth and stress-strain responses were found to be linear up to this strain, the elastic modulus is constant up to this value.

The average indentation depth applied to corneocytes in the AFM nanoindentation experiments was $h \approx 70$ nm. At this depth, for the AFM probe used, the strain was calculated using Eq. S29 to be ~ 1.16 . Since this strain is within the linear elastic region observed in the microindentation experiments, this justifies the use of the elastic modulus value obtained for PDMS from the flat punch experiments in the AFM tip calibration procedure; at least for depths relevant to the corneocyte measurements. Fig. S6 shows a typical micromanipulator loading curve and its associated stress-strain curve (obtained using Eqs S25 and S30) plotted up to a strain of 1.16 ($h \approx 15$ μm). This illustrates the linearly elastic behaviour of PDMS in the range of strains relevant to the AFM corneocyte data analysis, which uses a tip shape function derived using the associated elastic modulus calculated from the average slope of the fitted curves.

Fig. S6. (a) The force as a function of depth up to 15 μm for the indentation of PDMS with a flat punch using a micromanipulation technique. (b) The stress as a function of strain calculated from (a) using Eqs S25 and S30, respectively. The strain imposed at the maximum depth was 1.16.

5. *On previous concern 2: the authors said, “It is difficult to determine an accurate value of the tip radius of a sharp tip to enable the application of DMT or JKR”. Contrary, it is a fairly straightforward procedure using, for example, tip check samples. Moreover, it looks like this is exactly what the authors did to show the probe geometry in figure 1.*

Response: It is the case that we used tip check samples to obtain an initial estimate for the geometry of the probe. Unlike Dokukin and Sokolov ⁸, for their Hertz/DMT/JKR analyses, for example, we did not assume that the AFM tip may be approximated by a spherical cap for indentations smaller than the tip radius, since, as shown by the calibration performed using a reference elastomer, the tip has a complex geometry that can be characterised with an evolution of a power-law index, m , from 1.15 (for $h_c = 5$ nm) to 1.7 (for $h_c = 250$ nm). Thus, the DMT or JKR models could not be used. In summary, we believe the calibration method for a sharp probe that we adopted provides a more accurate measurement of the geometry compared with the tip check sample, which is consistent with the values of the Young’s moduli measured for the PMMA and also that a skin effect was not observed for the PMMA at least for the smallest indentation depth of 18 nm.

Furthermore, a Gwyddion model prediction of a PDMS force curve can be made using the profile data obtained from the tip check sample, rather than just the tip radius. This prediction is based on a polynomial fit of these profile data and using the value of the PDMS elastic modulus obtained from the micromanipulation experiments. The agreement between this prediction and a selection of experimental force curves shown in Fig. 1 is much poorer than that obtained using the polynomial tip radius function obtained from the average of the complete set of force curves. Both predictions are shown up to the maximum profile depth that could be obtained from the tip check sample, which was $h_c = 34$ nm. A closer agreement with the experiment data over the entire depth range using the Gwyddion tip calibration method would require a depth-dependent elastic modulus of a different value, which we do not believe is justified.

Figure 1. The measured (FC1–FC5) force as a function of depth for PDMS using a sharp AFM probe compared with the data calculated using the PDMS tip calibration approach and the Gwyddion tip check sample.

6. *On previous concern 4: the authors correctly recognized and calculated the lateral displacement error of the AFM cantilever. In the case of corneocytes, this error was 13 – 25% of the total penetration depth. However, it is unclear how they concluded that this large error results only in 6 – 9% in the calculation of the Young’s modulus. In the supplementary materials, in the caption to the supplementary table 3, they mentioned the work of Huang et al. However, that will work doesn’t seem applicable to the case of simple measurements done by the authors. Huang et al teach how to calculate the lateral deflection to compensate it using a specially developed AFM mode. Since it was not described in the manuscript, it is unlikely that the authors used that mode. Furthermore, to demonstrate the successful work of the methods, Huang et al used much longer cantilevers compared to the ones used by the authors. Such much longer cantilevers produce lower lateral displacement error. And even then, Huang et al concluded that the error in the modulus can reach 100%. So it is a complete mystery how the authors arrived in only 6 – 9% error. Without the special AFM mode with lateral compensation described by Huang et al, it does not seem practical to calculate the error of the Young’s modulus. To translate the lateral error into the change of the Young’s modulus without such a mode, one would need to know the Poisson ratio of the sample material and contact friction between the AFM probe and sample. Neither of them was mentioned by the authors.*

Response: We agree that we did not include sufficient information about the error calculation of the Young's modulus. It was based on the simple assumption that the unconstrained (i.e. no friction) lateral deflection calculated using the Huang et. al (2007)⁹ method led to an equivalent effective reduction in the vertical deflection of the cantilever once the effect of friction was included. This effective reduction during deflection is caused by parasitic distortion towards the end of the cantilever caused by lateral forces acting on the AFM tip (as shown in Figure 1 in ref. 9). We agree that the one-to-one assumption is too simplistic. We have therefore removed Table S3 (corneocyte Young's modulus) from the SI and edited Section S8 in an effort to address these concerns.

Applying the Huang method to the current cantilever geometry gives the ratio of the horizontal to vertical deflection to be 0.37, which is indeed greater than the value of 0.29 calculated by Huang (2007) for their cantilever example. We also confirm that we didn't use any special lateral deflection compensation AFM mode, which means that our indentation data are potentially subject to the errors of the type shown by Huang (Figure 4 in ref. 9).

For PDMS at the maximum indentation load the potential (unconstrained) horizontal deflection is less than 1% of the total penetration depth. Therefore, the effect of any lateral displacement on the AFM data for the elastomer was not significant, even considering the effect of friction (see Figure 7 in ref. 9 with Young's modulus $E = 2.8$ MPa and spring constant, $k = 28.5$ N/m).

For the case of corneocytes (Table 2, $E = 0.4$ – 1.2 GPa), the potential horizontal deflection was calculated to be in the range 13–25% of the total penetration depth and, consequently, is more significant. Nevertheless, in itself, this would be expected to have minor effects on the AFM corneocyte data since the change in the unconstrained cantilever end deflection angle caused by the maximum loads used in the current work is $\ll 0.1^\circ$. It is noted that the effects of the 10° mounting angle on the vertical component of the cantilever stiffness and the tip contact area are accounted for in the deflection sensitivity calibration procedure, and by using a reference sample of known elastic modulus.

The parasitic bending at the end of the cantilever as a result of friction will potentially lead to apparent reductions of the normal force and, hence, to underestimates in derived material property values for corneocytes. However, since the values of the elastic modulus obtained for PMMA using the current method are similar to those quoted in the literature (see Supplementary Section 6), it is unlikely that this leads to major systematic errors in the values given in Table 2 for cells, which have Young's modulus values between PDMS and PMMA. We therefore conclude that the current method gives sufficiently accurate values for the Young's modulus of corneocytes despite a special lateral deflection compensation AFM mode not being used.

Added to main text (line 298):

It is well known (see Supplementary Section 8) that there is a tendency for lateral deflection of an AFM probe during indentation that is intrinsic to the elastic deflection of a tilted cantilever beam with a finite tip length (see Figure 3). For the cantilever used in the current work, the lateral tip end deflection is estimated to be 37% of the normal deflection (see Supplementary Section 8). In practice, this percentage will be reduced by tangential contact forces that cause parasitic reductions in cantilever end deflection angles. This parasitic bending at the end of the cantilever will result in apparent reductions of the normal force and, hence, to underestimates in derived material property values. However, since the values of the elastic modulus obtained for PMMA are similar to that quoted in the literature (see Supplementary Section 6), it is unlikely that this leads to major systematic errors in the values given in Table 2 for cells attached to glass slides.

Corrected in SI:

S8 Lateral deflection of AFM probe tip

Huang et al.⁹ have noted that, during indentation experiments, the flexure of a tilted cantilever will tend to generate an undesirable horizontal (lateral) displacement to the AFM probe tip. They proposed that the horizontal tip end deflection of the probe, Δx , can be calculated as follows⁹:

$$\Delta x = \Delta x_1 + \Delta x_2 \quad \text{S31}$$

where Δx_1 is the horizontal translation of the tip base arising from the cantilever mounting angle, θ , and is given by:

$$\Delta x_1 = \Delta z \tan \theta \quad \text{S32}$$

where Δz is the vertical deflection of the probe tip arising from the flexure of the cantilever in response to the normal contact force. The vertical deflection is given by:

$$\Delta z = \frac{F}{k_c} \quad \text{S33}$$

where k_c is the AFM cantilever stiffness in the direction normal to the specimen surface.

Δx_2 is an additional component of the lateral deflection of the tip end caused by rotation of the tip base. This component is associated with the cantilever end deflection angle, $\Delta\theta$,

created in response to the normal force. For a tilted cantilever beam with a force applied at the free end, this is given by ⁹:

$$\Delta\theta = \frac{3\Delta z}{2L \cos \theta} \quad \text{S34}$$

where L is the AFM cantilever length. From simple geometry:

$$\Delta x_2 = \Delta\theta T \cos \theta \quad \text{S35}$$

where T is the AFM tip length. Hence:

$$\Delta x_2 = \frac{3T}{2L} \Delta z \quad \text{S36}$$

Therefore, from Eqs S31, S32 and S36, the ratio of the horizontal tip end deflection to the vertical deflection is given by ⁹:

$$\frac{\Delta x}{\Delta z} = \left(\tan \theta + \frac{3T}{2L} \right) \quad \text{S37}$$

For the cantilevers used in the current experiments, T and L are given by the supplier as 15 μm and 117 μm , respectively, and $\theta = 10^\circ$ (Nanowizard 4, Bruker, JPK BioAFM, Berlin, Germany). Hence, from Eq. S37, the ratio $\Delta x/\Delta z = 0.37$ for the current cantilever geometry.

The stiffness of the AFM cantilever used for the PDMS indentation experiments was $k_c = 28.5 \text{ N/m}$. Hence, at the maximum load of 250 nN (see S1D and Figure S1c), from Eqs S33 and S37, $\Delta z = 8.8 \text{ nm}$ and $\Delta x = 3.2 \text{ nm}$ respectively. The indentation depth at the maximum indentation load was typically $h = 445 \text{ nm}$ (S1D and Figure S1c). Hence, at the maximum load, the potential horizontal tip end deflection, Δx , is only 0.7% of the indentation depth, Δh . The shape of the loading curve is such the $\Delta x/\Delta h$ ratio will be even less than this at smaller loads. Therefore, the effect of any lateral displacement on the AFM data for the elastomer was not significant.

For the case of corneocytes, similar calculations gave $\Delta x/\Delta h$ in the range 13–25% and, consequently, the lateral deflection is a significant proportion of the indentation depth. Nevertheless, this would be expected to have minor effects on the AFM corneocyte data since the cantilever end deflection angle, $\Delta\theta$, caused by the loads used in the current work is still $< 0.03^\circ$ (Eqs S33 and S34 with $F_{max} = 1000 \text{ nN}$ for corneocytes on glass, see Fig. 3b). The effects of the 10° mounting angle itself on the vertical component of the cantilever stiffness and the tip contact area are accounted for in the deflection sensitivity calibration procedure, and by using a reference sample of known elastic modulus.

However, Eqs S32 and S36 are derived assuming that there is not a resistance to the lateral tip displacement and that the tip end is free to rotate during contact. In practice, tangential contact forces associated with friction and lateral material deformation will reduce the lateral tip deflection. These tangential forces will also produce parasitic reductions in the cantilever end deflection angles that can result in significant underestimations of the normal force⁹. The phenomenon of lateral displacement errors in AFM nanoindentation could be addressed in future studies using strategies such as active lateral motion compensation⁹, the use of non-deflecting cantilevers and reduced cantilever tilt angles¹⁰⁻¹².

7. *On previous concern 8: About humidity control for imaging of corneocytes. The authors added: "The measurements were done under controlled ambient conditions of 25.5°C and 35% relative humidity, which corresponds to a nominal dry state for these cells, comparable to the conditions on the skin surface in their native state." What is the "nominal dry state"? And if 35% humidity seems to be too low for the "native state"? References to support such an unusual statement are needed.*

Response: The term "nominal dry state" was employed to refer to corneocytes equilibrated under ambient conditions as opposed to the "dry state", which corresponds to zero (unbound) moisture at 0% RH. However, we agree that this nomenclature could be misleading, and we have changed it to "relatively dry state".

Corrected text (line 227):

The measurements were done under controlled ambient conditions of 25.5°C and 35% relative humidity (RH), which corresponds to a nominal dry state for these cells, comparable to the conditions on the skin surface in their native state.

8. *An additional concern/note: formula S35 is not applicable for the parameters used by the authors because the radius of the probe is smaller than the indentation depth. Therefore, it cannot be considered as indenting by a spherical probe. Strictly speaking, the authors cannot use the formula for the conical probe, S34 because the probe geometry is substantially different from a pure cone. The model for a conespherical indenter should be used.*

Response: The AFM tip cannot be considered to be either a pure cone or a parabola. Consequently, we have included the calculations for these two extreme cases since our probe has a complex geometry. In fact, as shown in Table S1, the power law index, m , varies from 1.1 to 1.7 with increasing depth, i.e., the probe can be approximated to a flat punch for shallow indentations, and to conical shape for greater indentation depths. For the indentation depths corresponding to those employed for the corneocytes, we consider that the approximation to a parabola is the closest geometry for which Eq. S35 (Eq. S39 in the revised SI) is applicable. The limitation of the indentation depth for a spherical probe does not apply to one that is parabolic.

1. Simões, B. D., *et al.* Rheological and Mechanical Properties of an Acrylic PSA. *Polymers* **15**, 3843 (2023).
2. Rheinlaender J, Schäffer TE. Mapping the creep compliance of living cells with scanning ion conductance microscopy reveals a subcellular correlation between stiffness and fluidity. *Nanoscale* **11**, 6982-6989 (2019).
3. MatWeb. *Overview of materials for Acrylic, Cast.*(18/03/2024).
4. Évora AS, Zhang Z, Johnson SA, Adams MJ. The effects of hydration on the topographical and mechanical properties of corneocytes. *JMBBM* **150**, 106296 (2024).
5. Riethmüller C. Assessing the skin barrier via corneocyte morphometry. *Exp Dermatol* **27**, 923-930 (2018).
6. Engebretsen KA, Bandier J, Kezic S, Riethmüller C, Heegaard NHH, Carlsen BC, Linneberg A, Johansen JD, Thyssen JP. Concentration of filaggrin monomers, its metabolites and corneocyte surface texture in individuals with a history of atopic dermatitis and controls. *J Eur Acad Dermatol Venereol* **32**, 796-804 (2018).
7. Riethmüller C, McAleer MA, Koppes SA, Abdayem R, Franz J, Haftek M, Campbell LE, MacCallum SF, McLean WHI, Irvine AD, Kezic S. Filaggrin breakdown products determine corneocyte conformation in patients with atopic dermatitis. *J Allergy Clin Immunol* **136**, 1573-1580.e2 (2015).
8. Dokukin ME, Sokolov I. On the measurements of rigidity modulus of soft materials in nanoindentation experiments at small depth. *Macromolecules* **45**, 4277-4288 (2012).
9. Huang L, Meyer C, Prater C. Eliminating lateral forces during AFM indentation. *Journal of Physics: Conference Series* **61**, 805 (2007).